# Inkjet-printed unclonable quantum dot fluorescent anti-counterfeiting labels with artificial intelligence authentication

Yang Liu[1], Fei Han[2], Fushan Li[1], Yan Zhao[1], Maosheng Chen[1], Zhongwei Xu[1], Xin Zheng[1], Hailong Hu [1], Jianmin Yao[1], Tailiang Guo[1], Wanzhen Lin[2], Yuanhui Zheng [2], Baogui You[3], Pai Liu[3], Yang Li[3] & Lei Qian[4]

An ideal anti-counterfeiting technique has to be inexpensive, mass-producible, non-destructive, unclonable and convenient for authentication. Although many anti-counterfeiting technologies have been developed, very few of them fulfill all the above requirements. Here we report a non-destructive, inkjet-printable, artificial intelligence (AI)-decodable and unclonable security label. The stochastic pinning points at the three-phase contact line of the ink droplets is crucial for the successful inkjet printing of the unclonable security labels. Upon the solvent evaporation, the three-phase contact lines are pinned around the pinning points, where the quantum dots in the ink droplets deposited on, forming physically unclonable flower-like patterns. By utilizing the RGB emission quantum dots, full-color fluorescence security labels can be produced. A convenient and reliable AI-based authentication strategy is developed, allowing for the fast authentication of the covert, unclonable flower-like dot patterns with different sharpness, brightness, rotations, amplifications and the mixture of these parameters.

[1] Institute of Optoelectronic Technology, Fuzhou University, Fuzhou 350116, China. [2] College of Chemistry, Fuzhou University, Fuzhou 350116, China. [3] Guangdong Poly Optoelectronics Co., Ltd, Jiangmen 529020, China. [4] TCL Corporate Research, No. 1001 Zhongshan Park Road, Nanshan District, Shenzhen 518067, China. [5] These authors contributed equally: Yang Liu, Fei Han. Correspondence and requests for materials should be addressed to F.L. (email: fsli@fzu.edu.cn) or to Y.Z. (email: yuanhui.zheng@fzu.edu.cn) or to L.Q. (email: qianlei@tcl.com)

Counterfeiting and forgery is a global problem that causes significant financial damage and poses security threats to individuals, companies, and society as a whole[1,2]. Over the past decades, counterfeit products have spread from daily consumer goods, to medicines and high-tech products[1]. Although majority of the products are protected by an anti-counterfeiting technique, the global economic loss of counterfeiting has been increasing annually and estimated to reach 1.7 trillion US dollars in 2015[3]. The reason for this is the currently used anti-counterfeiting technologies that rely on inkjet-printed security labels can be readily duplicated by counterfeiters due to their uniform patterns and predicable, deterministic decoding mechanisms[4–6]. Despite this, the inkjet-printing technique itself has many distinguished advantages in low production cost, mass production, material-effective utility, unlimited pattern design ability, and excellent compatibility with various ink materials as well as supporting substrates[7–12]. Furthermore, the inkjet-printed macroscale security labels allow fast, frequent authentication by the naked eyes or using a smart phone.

Security labels with physical unclonable functions (PUFs) could offer a practical solution to the limitation of the widely used anti-counterfeiting technologies and seem to be the most viable path to combat the increasingly serious global forgery issue[13]. A PUF is a physical object with an intrinsic, unique, random physical feature generated in a non-deterministic process[1,13–20]. The randomness characteristic of the feature guarantees unreplicable code outputs. To date, a significant progress in PUF encryption has been made in the anti-counterfeiting field, mainly focusing on generating random features composed of either rough surfaces[21,22] or discrete nanoparticle arrays[23–25] within the pre-defined pattern areas. For example, Bae et al. has demonstrated the use of randomly wrinkling silica-coated polymeric particles as PUF codes to produce unique artificial fingerprints for anti-counterfeiting applications[22]. Our group recently have utilized advantage of electrostatic self-assembly strategy to generate randomly arranged plasmonic (metal) arrays as PUF codes using fluorescein-doped silver-silica core-shell nanoparticles as building blocks, giving rise to multi-optical signal encoded, unclonable security labels[23]. However, the creation of well-defined two-dimensional (2D) graphic security labels that carry PUF codes requires the aid of expensive lithography techniques. Moreover, the decryption of such PUF-based security labels relies on a machine learning pattern recognition and comparison analysis that a real fingerprinting does[22]. The machine learning authentication technique only focuses on algorithms for pattern recognition and enhancement to extract the PUF codes.

In this work, we develop an inkjet-printable, artificial intelligence (AI) decodable, unclonable, fluorescence security label by combining the advantages of inkjet printing, portable smartphone microscopes and AI technique. The security labels with diverse pattern designs are fabricated through inkjet printing using II-VI semiconductor core-shell quantum dots as model ink. The surface decoration of print substrates, such as glass, plastic, or paper, with randomly distributed poly(methyl methacrylate) (PMMA) nanoparticles is critical for the successful inkjet printing of unclonable security labels. The polymer nanoparticles on the substrates are acted as stochastic pinning points at the three-phase contact lines of the ink droplets. Upon the solvent evaporation, the three-phase contact lines are pinned around the pinning points. The quantum dots in the ink droplets subsequently deposit on the pinning points, forming physically unclonable flower-like dot patterns (i.e., primary units for any 2D logos). The designed surface modification makes the security labels (i.e., 2D logos) unique and unclonable. By utilizing red, green and blue (RGB) emission semiconductor quantum dots, full-color pictures can be generated, which are invisible in the ambient environment. The fabricated security labels seem to be the same from batch to batch on the macroscopic level; however, at the microscopic level, they are quite different from each other. Compared with previous reports using the intrinsic surface topography of a material (e.g., scratch patterns, fiber weave, etc.) for PUF encoding[16], the system presented here has many advantages: (1) the developed printing strategy for the security label fabrication not only allows for various pattern design but also makes the mass-production at low cost possible; (2) the quantum dot ink is fluorescence active, guaranteeing the readout signals from suffering the interference by fingerprints and dusts; (3) the quantum dot security labels are only visible upon UV excitation, which offers the first layer of security, while the PUF nature originated from the random flower-like dot patterns provides the second more secure layer; (4) the first layer of security can be easily authenticated with naked eyes, while the second layer of security is able to be authenticated with AI technique rather than time-consuming machine learning algorithms. Moreover, we introduce AI (specifically deep learning) security label authentication concept and succeed in accurately and robustly decoding the unrepeatable flower-like dot patterns with different focusing degrees, brightness, rotation angles, amplification factors and the mixture of these parameters. The developed anti-counterfeiting technology meets all the requirements for commercial applications that are low cost, mass-producible, nondestructive, diverse full-color pattern design capability, unclonable, and convenient for authentication.

## Results

**Inkjet printing of unclonable quantum dot security labels.** The fabrication process of unclonable security labels by inkjet printing are illustrated in Fig. 1. Three types of commercially available II–VI semiconductor core-shell quantum dots (CdSe/CdS/CdZnS, ZnCdSe/CdZnS, and ZnCdS/CdZnS) that emitted red, green, and blue light were chosen as model inks owing to their high fluorescent quantum yield and outstanding stability (Fig. 1a). These quantum dots were synthesized using a well-established chemical route[26,27], in which high boiling point oleic acid and 1-octadecen were used as a capping agent and solvent, respectively (see Methods for details). Typical low-magnification transmission electron microscopy (TEM) and high-resolution TEM micrographs present an extremely narrow particle size distribution and high crystallinity of the synthesized quantum dots (Fig. 1a, b and Supplementary Fig. 1). The particle sizes of the red, green, and blue emission quantum dots are $6.5 \pm 2$ nm, $10.5 \pm 3$ nm, and $11.0 \pm 2.5$ nm, respectively. Generally, for quantum dots, the relationship between size and emissive wavelength accords with quantum confinement effect — that a small size corresponds to a larger band gap and emits short-wavelength fluorescence[28]. In this case, the emission wavelength of red, green and blue quantum dots shown in Fig. 1a was determined by the composition rather than the size of quantum dots (see Supplementary Fig. 1c)[26]. The Fourier-transform infrared absorption peaks at 2923 and 2854 cm$^{-1}$, corresponding to C–C and C–H stretching modes of the -CH$_2$- group, reveals the presence of oleic acid on the quantum dot surface (see Supplementary Fig. 2)[29]. The surface ligands are of importance for the preparation of highly dispersed and stable quantum dot inks as they can stabilize the quantum dots in non-polar solvents for months at room temperature. In this work, n-octylcyclohexane with low evaporation rate at ambient environment was adopted as model solvent for ink preparation. The low evaporation rate of the jetted ink droplet leads to a slow movement of the three-phase contact line on the print substrate, which allows us to dynamically study the quantum dot deposition behavior. Photographs of twenty jetted single

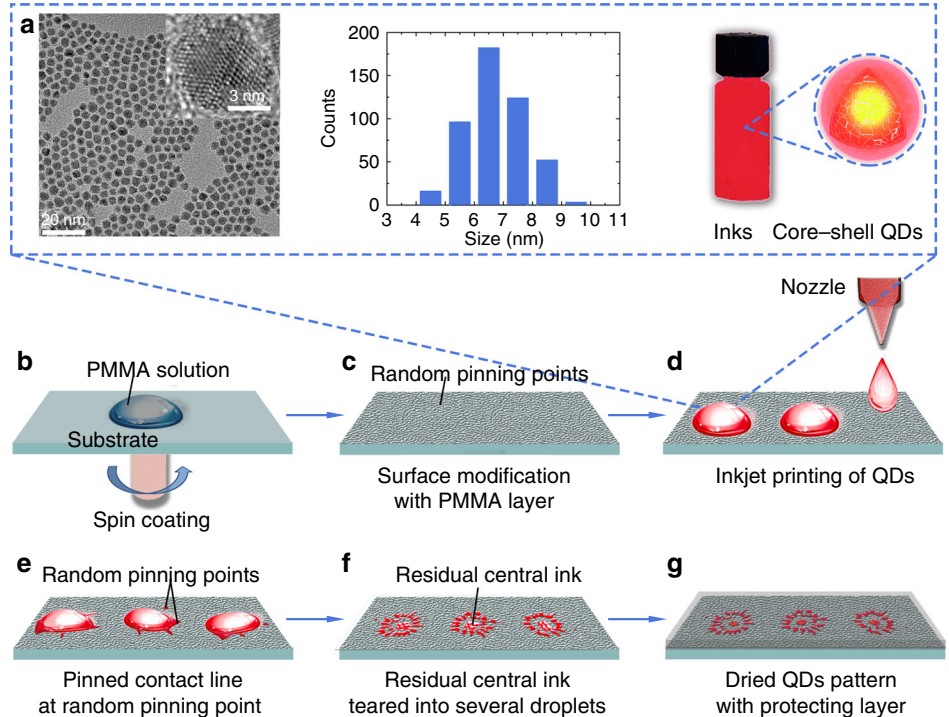

**Fig. 1** Outline of formation processes of unclonable security labels by inkjet printing. **a** Typical TEM/HRTEM images (left), particle size distributions (middle), and ink for inkjet printing (right) of red core-shell quantum dots (QDs). **b** Surface modification of a substrate with partly dissolved PMMA solution by spin coating. **c** Substrate with random-distributed pinning points of PMMA nanoparticles on PMMA film after spin coating. **d** Inkjet printing of prepared QDs ink on the modified substrate. **e** Pinning process of QDs droplet contact line at random pinning points. **f** Residual central ink teared in to several smaller droplets in the final evaporation process. **g** A resulting security label composed of flower-like dot patterns, which are protected with a thin, optically transparent sticky gel film

droplets show almost identical shape both before contacting a print substrate and at the early stage after the contact (see Supplementary Fig. 3).

Prior to inkjet printing, the print indium-tin-oxide (ITO)-coated glasses were cleaned by sonication in various solvents (see Methods for detailed sonication cleaning procedure), treated with oxygen plasma and then spin-coated with a partly-dissolved poly (methyl methacrylate) (PMMA)[30] nanoparticle colloid, forming a randomly distributed PMMA nanoparticle array on a PMMA layer (Fig. 1b, c). The surface cleaning and oxygen plasma treatment were utilized to remove the dusts and generate a super wettable surface for the PMMA nanoparticle coating, while the surface decoration with PMMA nanoparticles were used to create randomly arranged pinning points for the quantum dot deposition during the subsequent inkjet printing process. Such randomly arranged pinning points are critical for the successful inkjet printing of unclonable security labels (see Supplementary Fig. 6). Representative dark-field and atomic force microscopy images revealed the presence of randomly distributed PMMA nanoparticles with particle size in the range of a few nanometers to a few micrometers on the print substrates (see Supplementary Fig. 4). Following the surface modification of the print substrates, they were subjected to the inkjet printing to produce 2D graphic security labels. The inkjet printing was conducted by continuously jetting the quantum dot ink droplets with size of ~130 μm on the print substrates (Fig. 1d). The interval between two adjacent ink droplets is 200 μm. If each macroscopic security label contains 1000 flower-like points, we can achieve 1600 security labels in 5 min (i.e., the drying time of each batch of security label) with our single-nozzle printing machine. Each droplet represents a pixel of the inkjet-printed security labels after being completely dried. The PMMA nanoparticles on the poorly

wettable PMMA film were acted as stochastic pinning points at the three-phase contact lines of the ink droplets. With the evaporation of solvent, the three-phase contact line continually slides and shrinks at the smooth PMMA areas owing to the poor wettability of the surface but is captured by some pinning points. Consequently, the pinning points stretch and pin the contact line, distorting the fluid convex and forming irregular quantum dot ink pattern (Fig. 1e). During the solvent evaporation process, the quantum dot concentration increases gradually. When the quantum dot concentration reached their saturation point, they started to deposit at the stochastic PMMA pinning points on the three-phase contact line, forming a unique flower-like dot pattern (Fig. 1f). With the shrinking of droplet, the volume-smaller droplet is more liable to be tortured by pinning points, thus splitting into several smaller sub-droplets (see Supplementary Fig. 5). Due to the small space between these sub-droplets and some stochastic external factors (such as airflow), they experienced merging and splitting irregularly before eventual drying (Fig. 1f). This non-deterministic, random pattern formation process was recorded by a camera to monitor the shape evolution of a droplet as a function of time (see Supplementary Fig. 5). Such a non-deterministic pattern formation process makes it impossible to reproduce the flower-like dot patterns, which were used as PUF codes in our security labels. The fabricated security labels were then covered with a thin, optically transparent, sticky gel film to protect them being damaged during the real circulation (Fig. 1g). If without the PMMA nanoparticles on the print substrates, the accumulated quantum dots induced by solvent evaporation directly deposited on the self-pinned three-phase contact line, producing a coffee-ring-like pattern (see Supplementary Fig. 6a). However, if the print substrates were coated with a continuous, smooth, poorly wettable PMMA layer, the

three-phase contact line of the droplets are incline to shrink and slide towards to the center, rather than pinned on substrates, bringing the quantum dots to center of the droplet. When the solvent is completely evaporated, a central bump pattern, i.e., a hilly accumulation with diameter much smaller than the initial wetted diameter of the ink droplet, is generated (see Supplementary Fig. 6b). In the absence of the stochastic PMMA pinning points, either a uniform coffee ring or central bump pattern was formed, which was determined by the wettability property of the print substrates. These deterministic pattern formation processes ensure the reproducibility of the patterns under the same conditions.

The miniaturized dark-field and fluorescence microscopes integrated with smartphones have been developed for single nanoparticle imaging[31,32]. Such small, affordable, portable microscopes were utilized by consumers to authenticate the inkjet-printed security labels. The magnification-adjustable objective lens of the portable microscope used here is covered with a cylindrical metal shell that creates a small dark imaging environment by blocking the ambient light (Supplementary Fig. 7). The portable microscope is able to quickly (typically within one second), accurately and nondestructively readout the produced flower-like dot patterns, which show different geometries as expected (Supplementary Fig. 7).

**Unlimited colorful pattern design capability**. Ideal fluorescent security labels require the ink materials having strong fluorescence intensity, which guarantees the high fluorescence brightness. The prepared quantum dot inks exhibit a narrow fluorescence emission peak with maximum intensity at 459 nm, 532 nm, and 631 nm, respectively, when excited at 375 nm (Fig. 2a). The full-width at half-maximum (FWHM) of these blue, green and red emission quantum dots is 20 nm, 24 nm, and 28 nm, respectively. They also bear a high fluorescence quantum yield of 75, 90, and 80%, respectively. The time-resolved fluorescence decays of the blue, green, and red emission quantum dots show their lifetime of 27.5 ns, 17.0 ns, and 7.6 ns (Fig. 2b), respectively, in consistent with consistent with previous reports[26,33,34]. According to the Commission Internationale de I'éclairage (CIE) 1931 color coordinate triangle, the quantum dots achieved a wide color gamut of 124% of the National Television System Committee (NTSC) standard (see Supplementary Fig. 8), better than phosphor (color gamut: 85.6%)[35–37], implying outstanding full-color fluorescence properties for security label applications.

Complex Fuzhou University logos composed of red, green, or blue emission arrays were inkjet-printed on indium tin oxide (ITO) coated glass slides using the printing strategy described in Fig. 1. The print substrates with Fuzhou University logos maintained transparent (Insert of Fig. 2c). That is, the inkjet-

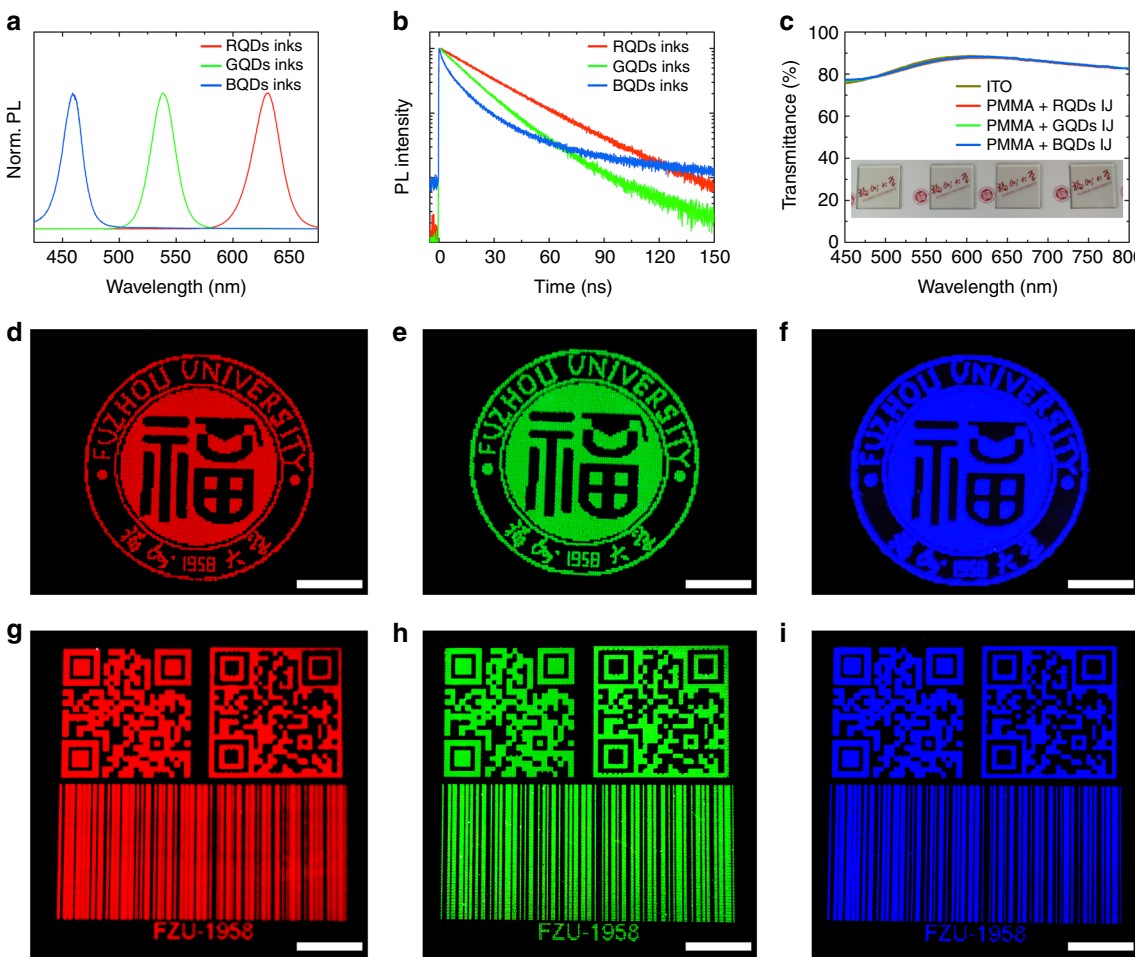

**Fig. 2** Optical characterization. **a** Fluorescence spectra and **b** time-resolved fluorescence decay measurements of red, green, and blue emission quantum dot (QD) inks excited at 375 nm. **c** Transmittance spectra of ITO coated glass substrates before and after inkjet printing of red, green, and blue emission Fuzhou University logos (Inset: photographs of the four samples under ambient light). **d**–**i** Fluorescence images of red, green, and blue emission **d**–**f** Fuzhou University logos and **g**–**i** two-dimensional QR codes and bar codes. The Fuzhou University logos, QR codes and bar codes, composed of thousands of dot patterns (pixels), are adapted with permission of the Fuzhou University. The scale bar is 1 cm

printed logos are invisible by naked eyes under the ambient conditions. The transmittance curves of the printed ITO glass slides are almost identical to the pristine print substrate (Fig. 2c), further confirming the covert characteristic of the fabricated security labels. When irradiated with UV light, bright red, green and blue photoluminescence photographs of the university logos were observed by the naked eyes (Fig. 2d–f). Any other 2D macroscale patterns with sharp corners, for example, QR codes and bar codes (Fig. 2g–i), can also be produced using our inkjet printing technique. The property that these images are only seen upon UV excitation offers the first layer of security realized by naked eye authentication of macroscopic patterns; and the PUF nature of the flower-like patterns is the second more secure layer.

By tuning RGB ink component ratios, fine tonal variations from red to green were obtained (see Supplementary Fig. 9). In principle, the full palette of colors could be achieved with elaborate control of the RGB ink components. Alternatively, the full-color security labels can be generated by creating individual color pixels including red, green, and blue sub-pixels, which has been widely used in display[26,38,39] and color images[40–42]. The inkjet printing strategy developed here offers us an opportunity to extend the ink to other fluorescent materials, e.g., lanthanide complexes[43–47], carbon dots[48,49], and up-conversion nanoparticles[50–52], etc. and the print substrates to flexible ones, such as plastics (see Supplementary Fig. 10). Although the 2D macroscale security labels shown in Fig. 2d–i contained thousands of discrete points (or pixels), the printing process is completed within a few minutes using our single-nozzle printing machine.

Figure 3 displays fluorescence micrographs of three batches of inkjet-printed macroscopic letters FZU (i.e., the acronym of Fuzhou University) that are composed of red, green, or blue emission arrays. At the macroscopic level, the printed FZU letters were seemingly identical (Fig. 3a–c). However, under the microscopic level, they were completely different from each other, as every fluorescent pixel of the letters showed a unique, random and unrepeatable flower-like geometry (Fig. 3d–i and Supplementary Fig. 11). Further scanning electron microscope and energy dispersive X-ray spectroscopy characterizations verify that the unique flower-like micropattern obtained from optical imaging systems is consistent with quantum dot deposition zones (see Supplementary Figs. 12–14). By carefully comparing all the pixels within the letters (i.e., the same sample), no identical flower-like micropatterns were found. A copy of the red, green, and blue fluorescence counterparts of the FZU letters shown in Fig. 3 fabricated under the same conditions also shows entirely different geometries of the corresponding pixels (see Supplementary Fig. 15). The encoding capacity of the inkjet-printed security labels that relies on the encoding capacity (defined as l) and the number (defined as m) of the PUF patterns (i.e., flower-like pixels) within the security labels can be described as $l^m$[20]. Recently, Carro-Temboury et al. established a universal binary-bit model for the estimation of the encoding capacity of PUF pattern[1]. According to Carro-Temboury's model, the encoding capacity of a red flower-like PUF pattern, l, is calculated to be $4.7 \times 10^{202}$ (see Supplementary Fig. 16 and Note 1 for calculation details). Therefore, for a security label composed of 1000 red flower-like PUF patterns, its encoding capacity will be larger than $10^{202,000}$. This indicates that the inkjet-printed security labels are unclonable even by the manufacturer. Moreover, the fabricated security labels have been frequently exposed to UV light over the period of 2 months. No obvious decrease of the fluorescence brightness is observed, revealing that security labels have excellent chemical and photo stabilities (see Supplementary Fig. 17).

**Deep learning decoding mechanism.** Conventional security labels with macroscale features, for instance, QR codes or bar codes are convenient for authentication with naked eyes, but easy to be counterfeited. Advanced security labels that carry complex, random nano/microscale features (PUF keys) are formidable to be faked. The decoding of such security labels relies on a similar character extraction, recognition, and comparison analysis that a real fingerprint does[22,23]. Such machine learning algorithms for pattern recognition, enhancement, and identification are widely employed for the authentication of the PUF-based security labels[22,23]. A general drawback of the machine learning approach is time consuming and having a high level of false positives of up to 20%[20]. Furthermore, conventional classification can only be

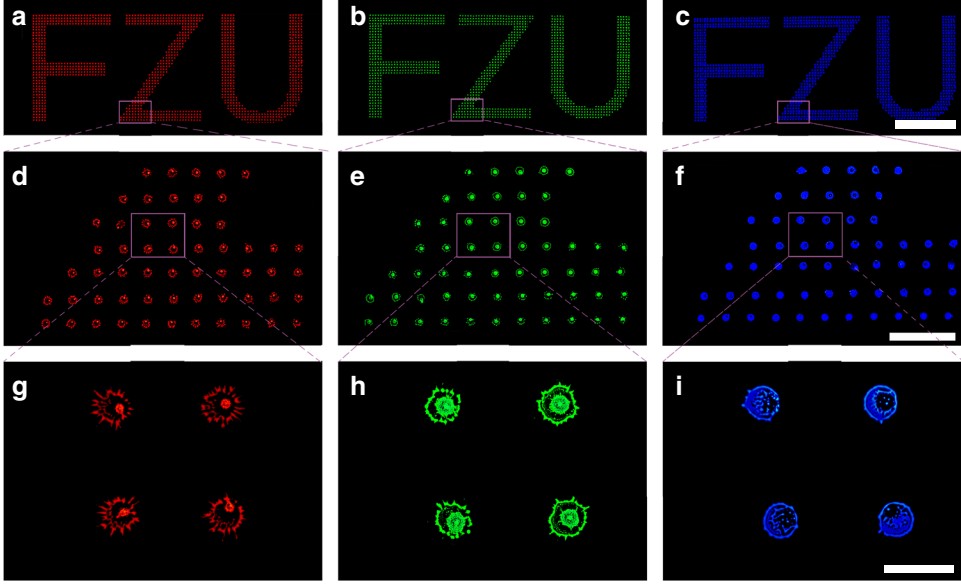

**Fig. 3** Security labels with physically unclonble flower-like patterns. Typical fluorescence images of inkjet-printed macroscopic letters FZU composed of **a** red, **b** green, and **c** blue emission dot patterns (the scale bar is 4000 μm), **d–f** their local enlarged images (the scale bar is 500 μm), and **g–i** further enlarged images (the scale bar is 100 μm). All the samples are printed under the same conditions. The FZU patterns are identical on macroscale but completely different at the microscale (i.e., every dot pattern is unique)

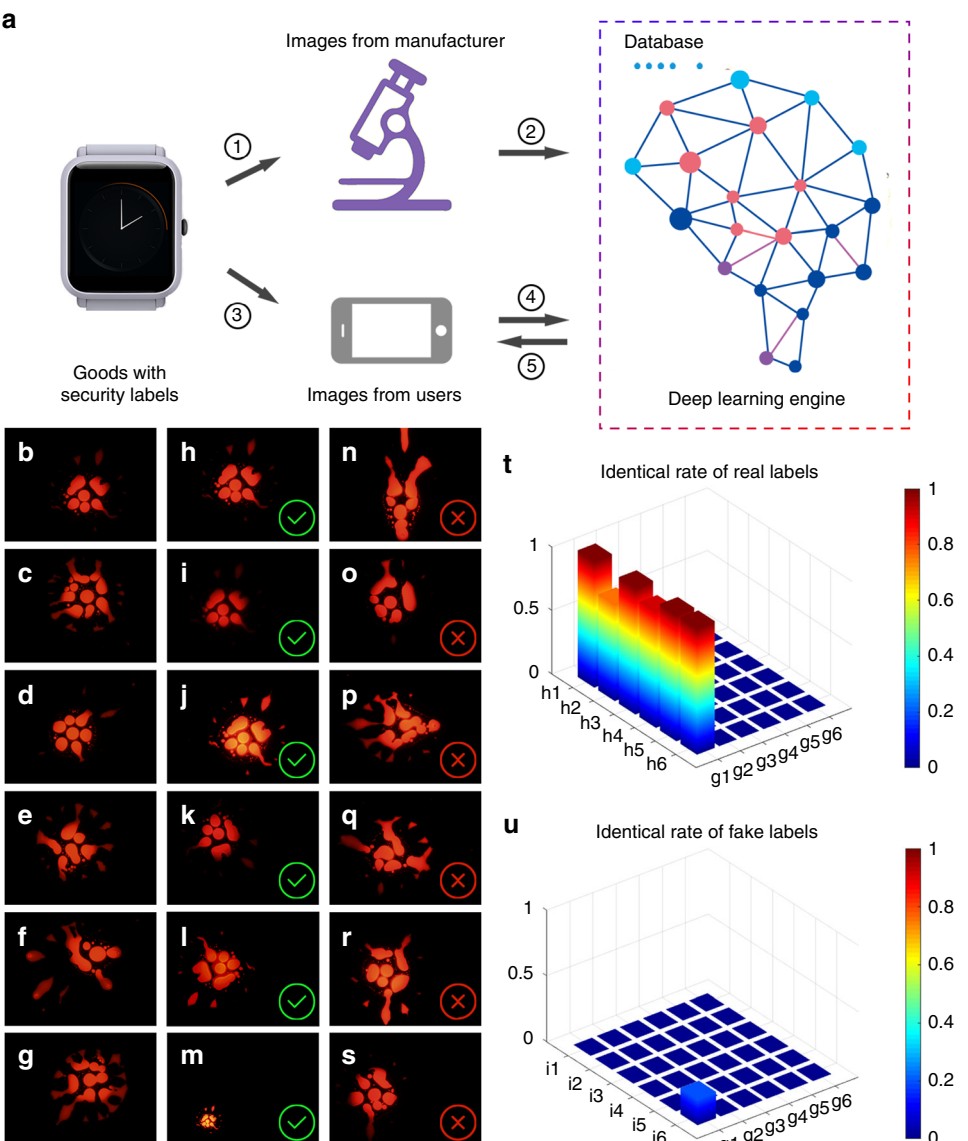

**Fig. 4** Deep learning decoding mechanism. **a** Schematic illustrating the authentication process: 1: image capture and database generation from the goods by the manufacturers, 2: image learning by AI, 3: Image capture from the goods by the consumers using their smart phones, 4: image recognition and comparing by AI, and 5: authentication outcome feedback to the consumers. **b–g** A library of six single dot pattern security labels, **h–m** six fluorescence images taken from **b** (referred as to genuine product) with different brightness, sharpness, rotation angles, magnifications, and the mixture of the above-mentioned factors, and **n–s** six fluorescence images from security labels that are not in the database shown in panel **b–g** (referred as to fake products). **t**, **u** Recognition rates of labels (h1–h6) and (i1–i6) by the authenticating way of deep learning. Labels (g1–g6), (h1–h6), and (i1–i6) correspond to the labels in figure (**b–g**), (**h–m**), and (**n–s**) respectively. The color scales from blue to red stand for the matching score (ranging from 0 to 100%) of the captured pictures with the labels

used in the PUF that can be transformed to private keys; for flower-like patterns, it does not work. To deal with this problem, deep learning, an artificial intelligence (AI) technique, was introduced to validate the fabricated unclonable security labels, which pushes anti-counterfeiting technology to a higher dimension.

Although AI has been recently applied into organic chemistry synthesis and TEM image analysis[53,54], to the best of our knowledge, this is the first time to use AI for security label authentication. Figure 4 demonstrates a typical authentication procedure of the inkjet-printed security labels through deep learning. First of all, each quantum dot security label on a commercial product is imaged using an advanced fluorescence microscope (Fig. 4a, step 1). Only one image is taken from each security label and represents one PUF code. Simply, randomly

shifting and rotating the image creates a large number of images for an AI to learn the characteristic features of the security label. Once the images are trained on AI, they are categorized in a very general manner (e.g., classes associated with geometry of the security labels) and then stored as a database in a deep learning engine for the subsequent authentication using (Fig. 4a, step 2). This is done by the manufacturer. When the consumers receive the products, they can simply use their portable mini-microscope-connected smartphones to readout the PUF codes by taking photos of the security labels (Fig. 4a, step 3), which are automatically sent to the deep learning engine for validation (Fig. 4a, step 4). The deep learn engine immediately feeds back the authentication results (real or fake) to the users (Fig. 4a, step 5).

To experimentally demonstrate the above authentication process, six quantum-dot security labels (named as $g_n$, $n = 1, 2,$

···, 6) were randomly chosen to establish a security label database (Fig. 4b–g). Five hundred fluorescence images of each security label (e.g., $g_1$) obtained by randomly shifting and rotating a same image (i.e., $g_1$) are provided to AI for learning and classifying. The selected 72 out of the 500 images from $g_1$ show the exact same geometrical characteristics of the security label (see Supplementary Fig. 18). The 500 images were divided into two parts: 80% for learning and 20% for validation. After every learning cycle (the parts for learning), the images for validation were sent to AI engine to test, providing a train accuracy plot. After about 1000 learning cycles, they can be recognized with accuracy fluctuating between 97 and 100% when being sent to AI for validation again (see Supplementary Fig. 19).

For decoding, a security label that represented a genuine product (i.e., from the pre-established database) was imaged at various sample rotation angles, magnification, focusing degrees, and the mixture of the above-mentioned factors (Fig. 4h–m). We try to cover all possible deviations from the imaging equipment, imaging conditions, personnel habits of users that may happen in a real authentication scenario. None of the images shown in Fig. 4h–m (named as $h_n$, $n = 1, 2, \cdots, 6$) has ever been previously learnt by AI. The image $h_n$ is fed into the trained AI for validation. The authentication outputs show the accuracy of $h_n$ ($n = 1, 2, \cdots, 6$) is 0.999, 0.758, 0.999, 0.909, 0.999, and 0.999, respectively (Fig. 4t). The relatively low accuracy of $h_2$ and $h_4$ are attributed to their over-indistinct characteristics, for which parts of details are lost during imaging, implying sharpness exerting a much higher impact on authentication accuracy than other variations, such as brightness, location, rotation angle, and amplification factor. The threshold of the accuracy at a value of 0.5 is then set to distinguish the real and fake security labels. For comparison, six fake security labels named as $i_n$ ($n = 1, 2, \cdots, 6$) were sent to AI for authentication in the same way (Fig. 4n–s). The corresponding accuracy is almost zero for all the fake security labels (Fig. 4u). By simply comparing the accuracy on the test with the threshold, the deep learning machine can immediately provide the authentication outcomes (real: accuracy ≥0.5, fake: accuracy <0.5) to the customers. Regarding to the rate of false positives, we achieved the false positives rate of 0 using the match score of 0.5 as the threshold when sampling 100 security labels (see Supplementary Table 1). It only takes seconds or even less to finish the whole authentication process.

## Discussion

In conclusion, we have demonstrated a non-destructive, inkjet-printable, smart-phone readable, AI decodable, unclonable, and fluorescence security label. Such security labels with various 2D patterns composed of red, green, or blue emission arrays were fabricated through inkjet printing using II–VI semiconductor core-shell quantum dots as model ink. The surface modification of the print substrates with randomly arranged PMMA nanoparticles is crucial for the successful inkjet printing of the unclonable security labels. The polymer nanoparticles on the print substrates were acted as stochastic pinning points at the three-phase contact lines of the ink droplets for the quantum dot deposition, forming physically unclonable flower-like dot patterns. This non-deterministic pattern formation process guarantees the unclonability of the fabricated security labels. By utilizing the RGB emission core-shell quantum dots, full-color security labels can be generated.

The fabricated security labels are invisible in the ambient environment but can be visualized by naked eyes when irradiated with UV light, which offers an easy way for the preliminary verification. A more reliable authentication strategy by using AI techniques has been developed. Covert and unclonable flower-like

dot patterns with different sharpness, brightness, rotations, amplifications, and the mixture of these parameters have been successfully decoded within seconds using the authentication strategy developed here. The overall cost per security labels has been estimated to be approximately US$ 0.011 (see Supplementary Note 2). The anti-counterfeiting technology described in this work is a good step closer to commercial applications that are low cost, mass-producible, nondestructive, diverse full-color pattern design capability, unclonable, and convenient for authentication. Applications of this technology span the full range from established to emerging-technology industries, including pharmaceutics, food security, and nanotechnology. Compared with intrinsic surface topography of material itself (like scratch patterns, fiber weave, etc.), our current system shows advantages in fluorescent and multi-color information, multi-level security, convenient for authentication, and well-designed patterns. The possibility of covert but easily detectable labeling through the use of a portable mini microscope and AI technique (Supplementary Table 2) will ensure the security and trackability of sensitive substances and equipment, which will lead to a promising approach in sensitive industries such as the nuclear one.

## Methods

**Materials**. Poly(methyl methacrylate) (PMMA, average Mw ~996,000 g mol$^{-1}$, from Sigma-Aldrich), polyvinylpyrrolidone (PVP, average Mw ~40,000×$g$ mol$^{-1}$), dimethyl sulfoxide (DMSO), chlorobenzene (99.8% pure, from J&K Scientific), CdO (AR grade, from Aladdin), ZnO (99.7%, from Shijiazhuang hongda zinc industry co. LTD), zinc acetate dehydrate (AR grade, from Aladdin), oleic acid (OA, 90%, from Alfa aeser), 1-octadecene (ODE, 98%, from Toyata), Se powder (99.999%, from Alfa aeser), sulfur powder (99.95%, from Aladdin), 1-Dodecanethiol (DDT, ≥ 98%, from Chevron Phillips Chemical), toluene (for synthesizing quantum dots, AR grade, from Guangdong Guanghua Sci-Tech Co., Ltd), and ethanol (AR grade, from Guangdong Guanghua Sci-Tech Co., Ltd) were received and used without further purification. Sticky-gel film with gel thickness of 1.5 mm and retention level of X4 was purchased from Gel-Pak company. Toluene (for dissolving PMMA) was purchased from Sinopharm Chemical Reagents and further dried by distillation over sodium.

**Synthesis of core-shell quantum dots**. For a typical synthesis of CdSe/CdS/CdZnS red emission quantum dots: 7 mmol of CdO, 10 mL of OA and 25 mL of ODE were mixed in a 250 mL round flask. The mixture was heated to 160 °C, degassed for 15 min, then filled with N$_2$ gas and further heated to 310 °C. Subsequently, Se precursor was injected swiftly into the flask. Then the reaction was cooled to 300 °C and remained at this temperature for 5 min. To grow the CdS shell, S source was injected dropwise to react with the remaining cadmium ions for 20 min. For the growth of CdZnS shell, keep the reaction temperature and drop-wise injected the pre-prepared shelling materials, the shelling process last for ~30 mins. After the reaction, the temperature was naturally cooled down to room temperature. The synthesized quantum dots were finally purified using toluene and ethanol for several times and finally dispersed in n-octylcyclohexane. For ZnCdSe/CdZnS green emission quantum dots: 35 mmol of ZnO, 25 mL of OA, and 20 mL of ODE were mixed in a 250 mL round flask. The mixture was heated to 160 °C, degassed for 15 min, then filled with N$_2$ gas and further heated to 300 g to get a clear solution. The solution was cooled to 200 °C, at which Se and Cd stock precursors were quickly injected into the flask sequentially. Then the temperature was elevated to 310 °C and remained at this temperature for 30 min to form alloyed CdZnSe core. For the growth of the CdZnS shell, S source and Cd source were dropwise injected into the flask repeatedly while keeping temperature at 270 °C. After the precursor injection, the temperature was kept unchanged for 30 min to 1 h, and then naturally cooled down to room temperature. For the synthesis ZnCdS/CdZnS blue emission quantum dots with composition gradient, the procedure is the same as that for the green emission quantum dots, except using S stock solution instead of Se solution as the anion precursor. At the elevated temperature, the S precursor was injected into the mixture of the Zn precursor and Cd precursor in a round bottom flask, forming the blue emitting ZnCdS cores. The shell materials (S source and Cd source) were added afterwards drop-wisely, and the reaction was allowed to proceed from 30 min to 1 h for the shell growth. The synthesized of green and blue quantum dots were purified and prepared with similar process as above.

**Fabrication of quantum-dot security labels**. In a typical procedure, the precursors of surface modification layer were prepared by adding solutes into chlorobenzene, toluene, and DMSO for common PMMA (4 mg ml$^{-1}$), designed PMMA (4 mg ml$^{-1}$), and PVP (4 mg ml$^{-1}$) layer respectively, and stirring vigorously for 12 h at 60 °C before use The quantum dot inks were dispersed in n-

octylcyclohexane at a concentration of 20 mg ml$^{-1}$. Indium-tin-oxide (ITO)-coated glass substrates were cleaned with ultrasonication successively in deionized (DI) water, acetone, isopropanol, and DI water. Then, nitrogen stream was used to dry the substrates, followed by 10 min oxygen plasma treatment. The pre-prepared precursors of surface modification layer were deposited on the substrates by spin coating, and then were heated at 120 °C for 30 min. The quantum-dot inks were printed on the substrates with a Microfab JETLAB II equipped with a 30-μm diameter piezoelectric-driven inkjet nozzle and a motorized stage with the accuracy of 5 μm. Driving voltage waveforms to the inkjet printing nozzle and the flying single droplet are shown in Supplementary Figs. 20 and 21. All the processes were operated in the ambient environment. If every single macroscopic pattern has 1000 unclonable flower-like points, we can achieve 1600 macroscopic patterns in 5 min with our single-nozzle printing machine. The time of drying is ~5 min for each batch of security label. The as-fabricated security labels were covered with gel films by tearing of their polycarbonate coversheet and then stick their gel material on the labels for stability test (see Supplementary Fig. 17).

**Characterization**. The surface morphology of the PMMA films and quantum-dot patterns were characterized with atomic force microscopy (AFM, Bruker Multimode 8) and scanning electronic microscope (SEM, FEI, Nova Nano SEM 230). The UV−Vis absorption spectra were tested with a UV/Vis/NIR spectrophotometer (Shimadzu, UV-3600). The transmission electronic microscopy (TEM) image of quantum dots was recorded using JEOL JEM-2100F microscope. The steady-state photoluminescence (PL) spectra were collected with a Hitachi F-4600 fluorescence spectrophotometer, by exciting the samples using a Xe lamp coupled with a monochromator. Time-resolved PL measurement was collected by using fluorescence lifetime measurement system (HORIBA scientific). Fourier transform infrared (FTIR) spectra were recorded with a Nicolet 50 FTIR spectrometer at room temperature. The PL microscopic images of quantum dot patterns morphology were characterized by using a fluorescent microscope (Olympus BX51M). For pattern readout with a smartphone microscope, the portable microscope was linked to a small WiFi box by a USB line, which allows for the smartphone to control the microscope for real-time imaging (see Supplementary Fig. 7).

**Quantum yield of quantum dots**. The results are obtained by comparing integrated PL intensities using the standard procedure[55,56]. The quantum yield (QYs) of blue, green, and red emission quantum dots were measured relative to Coumarin 480 (QY 99% in ethanol) with excitation at 350 nm, Coumarin 480 (QY 99% in ethanol) with excitation at 370 nm and rhodamine 6 G (QY 95% in ethanol) with excitation at 450 nm, respectively. Solutions of quantum dots in toluene were optically matched at the excitation wavelength. Fluorescence spectra of quantum dots and dye were taken under identical spectrometer conditions in triplicate and averaged. The optical density was kept below 0.06 at the $\lambda_{max}$, and the integrated intensities of the emission spectra, corrected for differences in index of refraction and concentration, were used to calculate the quantum yields using the expression.

$$\text{QY of quantum dots} = \text{QY}_R \times \frac{I}{I_R} \times \frac{A_R}{A} \times \frac{n^2}{n_R^2} \quad (1)$$

where QY is the quantum yield, $I$ is the measured integrated PL emission intensity, $n$ is refractive index ($n = 1.496$ for toluene; $n = 1.361$ for ethanol) and $A$ is the optical density at the excitation wavelength.

**Deep learning**. All the deep learning networks employed in our paper are based on the TensorFlow backend. The code used are run in software Pycharm 2017.3. To generate a large enough training set from theoretical image(s) for Alexnet model, a data augmentation procedure to the original synthetic image(s) is applied. For a typical process, one lower-left dot representing a security label is captured as an image. Such a clear image was rotated by a step of 0.72° for 360° using an algorithm, producing a set of 500 training images. The input images were resized to $512 \times 384$ using pixel area relation for training. Plots of accuracy on the training and validation data sets over training epochs (from http://host:6006) can be found in the Supplementary Fig. 19. The training images do not need to be stored and are not stored in this case; The storage requirement is mainly determined by the neural network itself, about 200 M Bytes here. For the AI technology we used, the learning process takes 2 h. The computer used for deep CPU is equipped with the CPU (Intel(R) Core(TM) i7–6700 CPU @3040 GHz), the GPU (NVIDIA GTX 1080), the RAM (32.0 GB), and HDD Capability (1 TB). The computer rated power is 350 W/h.

**Registration and validation methodology**. We created a file named as gn ($n = 1$, 2, 3, …) per security label to store the corresponding 500 training images prior to the training process. The training images stored in the file gn are named as gn_000, gn_001, …, gn_500. Many files from these security labels composed a database. After the 500 training images of a security label were learnt by AI, their structural information was remembered and linked to the file name gn (e.g., g1). Then the training images will be deleted. When consumers randomly take a picture of a real security label and sent it to the AI, the AI can automatically recall the accurately corresponding relationship and output the indexing name with a detailed match score. According our results, if the captured image from the end-user is clear

enough, the match score of the image of true security label is more than 99% (Fig. 4t). On the other hand, if the image (from a fake label) has never been learned, the engine will give a lower match score (Fig. 4u). The authentication process takes about 2 s. Deep learning, as a black box that nobody actually knows how it works in details up to now, is an advantage for unclonable anti-counterfeiting technique because it is a tamper proof.

## Data availability
All relevant data supporting the findings of this study are available from the corresponding authors on request.

## Code availability
AlexNet, a convolutional neural network competed in the ImageNet Large Scale Visual Recognition Challenge[57] was used here. The code can be assessed at https://github.com/deep-diver/AlexNet.

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

## Acknowledgements

We acknowledge the generous financial support from the National Natural Science Foundation of China (Grant No. U1605244, 61605028, and 61775040), the National Key Research and Development Program of China (Grant No. 2016YFB0401305), Program for Thousand Young Talent plan and Guangdong Provincial Science and Technology project (Grant NO. 2016B090906001). We acknowledge the help of Jiangmen Innovative & Entepreneurial Research Team Program in synthesizing quantum dots.

## Author contributions

F.S.L. and Y.H.Z. conceived the project. Y.Liu and F.S.L. designed the experiments and fabricated the devices. Y.Liu and F.H. collected and analysed the data. M.S.C., Z.W.X., X.Z. H.L.H., T.L.G. and W.Z.L. assisted sample characterization and data analysis. Y.Liu, Y.Z. and J.M.Y conducted deep learning authentication. B.G.Y., P.L. and Y.Li from Guangdong Poly Optoelectronics Co., Ltd offered help in synthesizing the quantum dots. Y.Liu, F.H., F.S.L., Y.H.Z. and L.Q. wrote the manuscript. All the authors read and commented on the paper.

## Additional information

**Competing interests:** The authors declare no competing interests.

