## [Peer Review File · Nature Communications]

Reviewers' comments:

Reviewer #1 (Remarks to the Author):

This manuscript is the first example I have seen where physical unclonable functions have been used in combination with machine learning to create an optical authentication system. The PUF is created by printing a luminescent ink on a hydrophobic substrate that contains random defects (random ordering) such that the pattern created when the ink dries can be considered a PUF. The only drawback of the presented method is the need to use a fluorescent microscope to see the authentication pattern. The end-user will not such a microscope, and simple authentication using a smart-phone and e.g. a lens is not possible as the ambient light will disturb the reading of the luminescent PUF patterns.

I find that the work and the approach is interesting and can be published if the authors address the three major problems I have identified in their work:

1) The authentication system must be tested in a way that can provide an estimate of the encoding capacity (should be 10^{20} or larger), and the rate of false positives. These two numbers, the latter in particular as it dictates the level of security offered by the system.

2) The registration and validation methodology is not reported at a sufficient level of detail. For instance: How are the images indexed, and how is this index read and reported to the matching software? A search of all images to find a match and then report a positive results is probably not how this is done? Are the training images stored? If they are, what is the storage requirement for an individual PUF tag? How fast is the learning protocol? How many spots are used from each label and how are they selected? A fully descriptive walk-through of the procedure is needed. Considering the time- and storage requirement of each step and all inputs needed for each operation.

3) The authors have not cited the landmark reports in the field. There are several important papers that are not mentioned, these must be included, these 5 papers are listed below. There are several other papers on PUFs that could also be included, these are also listed below.

These references needs to be read and cited:

Prior to publication these references must be included as they are the first examples of the use of PUFs for authentication.

- Horstmeyer, R.; Judkewitz, B.; Vellekoop, I. M.; Assaworarratit, S.; Yang, C., Physical key-protected one-time pad. *Sci. Rep.* 2013, 3, 3543.

- Takahashi, T.; Kudo, Y.; Ishiyama, R. In *Mass-produced Parts Traceability System Based on Automated Scanning of "Fingerprint of Things"*, Fifteenth IAPR International Conference on Machine Vision Applications (MVA), Nagoya University, Nagoya, Japan, Nagoya University, Nagoya, Japan, 2017.

- Carro-Temboury, M. R.; Arppe, R.; Vosch, T.; Sørensen, T. J., An optical authentication system based on imaging of excitation-selected lanthanide luminescence. *Science Advances* 2018, 4 (1), e1701384.

-P. Long, Y. Y. Feng, C. Cao, Y. Li, J. K. Han, S. W. Li, C. Peng, Z. Y. Li, W. Feng, *Adv. Funct. Mater.* 2018, 28, 1800791.

- Wigger, B.; Meissner, T.; Forste, A.; Jetter, V.; Zimmermann, A., Using unique surface patterns of injection moulded plastic components as an image based Physical Unclonable Function for secure component identification. *Scientific reports* 2018, 8 (1), 4738.

The authors are strongly encouraged to also read and include these references:

- Bae, H. J.; Bae, S.; Park, C.; Han, S.; Kim, J.; Kim, L. N.; Kim, K.; Song, S.-H.; Park, W.; Kwon, S., Biomimetic microfingerprints for anti-counterfeiting strategies. *Adv. Mater.* 2015, 27 (12), 2083-2089.
- Smith, A. F.; Patton, P.; Skrabalak, S. E., Plasmonic nanoparticles as a physically unclonable function for responsive anti-counterfeit nanofingerprints. *Adv. Funct. Mater.* 2016, 26 (9), 1315-1321.
- Geng, Y.; Noh, J.; Drevensek-Olenik, I.; Rupp, R.; Lenzini, G.; Lagerwall, J. P. F., High-fidelity spherical cholesteric liquid crystal Bragg reflectors generating unclonable patterns for secure authentication. *Sci. Rep.* 2016, 6, 26840.
- Hu, Z.; Comeras, J. M. M. L.; Park, H.; Tang, J.; Afzali, A.; Tulevski, G. S.; Hannon, J. B.; Liehr, M.; Han, S.-J., Physically unclonable cryptographic primitives using self-assembled carbon nanotubes. *Nature Nanotechnol.* 2016, 11 (6), 559-565.
- Kim, J.; Yun, J. M.; Jung, J.; Song, H.; Kim, J.-B.; Ihee, H., Anti-counterfeit nanoscale fingerprints based on randomly distributed nanowires. *Nanotechnology* 2014, 25 (15), 155303.
- Tian, L.; Liu, K.-K.; Fei, M.; Tadepalli, S.; Cao, S.; Geldmeier, J. A.; Tsukruk, V. V.; Singamaneni, S., Plasmonic nanogels for unclonable optical tagging. *ACS Appl. Mater. Interfaces* 2016, 8 (6), 4031-4041.
- Herder, C.; Yu, M. D.; Koushanfar, F.; Devadas, S., Physical unclonable functions and applications: A tutorial. *Proc. IEEE* 2014, 102 (8), 1126-1141.

Please also correct the following minor issues.

The spots in figure 3 and 4 are clearly different in shape and form, please comment.

Line 20: The method is not smartphone readable, must be deleted. It is read using a fluorescent microscope attachment connected to a smartphone

Line 21: nothing is unlimited, a number must be reported, see also 1) above.

Line 30: not read using a smartphone, please delete

Line 30: for the AI technique please comment on the demand of computer power and storage space

Line 56: Sentence must be referenced, use e.g. Arppe et al.

Line 57: Sentence must be referenced, use e.g. Arppe et al.

Line 62: Sentence must be referenced, I am not aware of a reference that can support the claim.

Line 66: Naked eye authentication is not mentioned below, please delete.

Line 69: Sentence must be referenced.

Line 70: Sentence need more references, use e.g. Arppe et al., Carro-Temboury et al. and Takahashi et al.

Line 82: Sentence must be referenced.

Line 96: This is a postulate that needs to be substantiated

Line 99: the claim that they are quite different needs to be documented and reported as a number

e.g. rate of false positives or encoding capacity.

Line 105: The method is not convenient for authentication. First it needs a microscope. And second, no time for authentication is given and no detailed method of registration and validation is reported.

Line 112: Oil-phase route, is this the right word?

Line 118: I cannot decipher this sentence, please rephrase

Line 120: sentence/claim must be referenced

Line 122: sentence/claim must be referenced

Line 133: Please document that the method works on different substrates by including images in the SI, otherwise delete.

Line 134 various solvents, either specify or refer to methods section

Lines 140-141: undocumented claim, please document.

Lines 149-152: The time of printing and drying is critical to mass-production. How fast is the total process?

Line 157: I cannot decipher this sentence, please rephrase

Line 165: Sticky-gel film? Does not make sense, please give details and exact procedure to laminate codes in Methods section.

Line 179-180: Please mention the issue of have consumers use this type of equipment, and be critical as it is a major issue.

Lines 188-189: Please supply all data for QY determination as SI

Lines 203-207: Please make sure to state that all these images are only seen upon UV excitation. It is a good feature as that is the first layer of security, the PUF nature is the second more secure layer.

Line 211: Please cite the lamnthanide complexes used in euro bank notes. Andres et al Adv. Mater.

Lines 214-215: The procedure described here cannot be mass produced. Please comment.

Line 231: The encoding capacity is not infinite. Please consult the detailed considerations in the supporting information of Carro-Temboury et al.

Line 240: Sentence must be referenced.

Line 252: The number and types of characteristic features must be given and described.

Line 256: IT is not just a smartphone, delete or rephrase.

Line 258: How long does authentication take, how is it done, and how many can be run in parallel? This is critical.

Line 264: Which exact same geometrical characteristics?

Line 268: A number close to 1 can be many things, please be more concrete.

Lines 276-283: Please use the actual match scores, the threshold values and the non-match scores to calculate the rate of false positives and the actual encoding capacity. The latter must be a function of the threshold value.

Line 286: Time claim must be validated.

Line 300: Again. A smartphone is not needed, you need a microscope.

Line 305: This is not true. There is not an easy authentication without the use of specialised equipment and the printing method/drying time is not compatible with modern means of mass production. And neither is ITO and spin-casting. It is a good step closer to a compatible method, but we are not there yet.

Lines 324-: Is stirring used at all?

Line 378: the symbol before 500 is missing

Line 310: Again. A smartphone is not needed, you need a microscope.

Reviewer #2 (Remarks to the Author):

Author reported a non-destructive, inkjet-printable, smart-phone readable, AI decodable and unclonable security label. In this process, author addressed 1) before inkjet printing, the print substrate with random-distributed pinning points of PMMA nanoparticles on PMMA film is used. 2) 2D patterned security labels composed of red, green or blue arrays are fabricated through inkjet printing using II-VI semiconductor core-shell quantum dots as inks. 3) forming physically unclonable "flower-like" patterns using the creation of stochastic pinning points at the three-phase contact line of the ink droplets. 4) authentication is done by deep learning decoding mechanism. Five hundred fluorescence images of each security label obtained by randomly shifting and rotating a same image are provided to AI for learning and classifying. The threshold of the accuracy at a value of 0.5 is then set to distinguish the real and fake security labels. For comparison, six fake security labels were sent to AI for authentication then the corresponding accuracy is almost zero for all the fake security labels. Author conclude that the inkjet-printing technique guarantees the mass production of security labels at low cost and the developed authentication strategy allows for the fast authentication of the covert, unclonable "flower-like" dot patterns with different sharpness, brightness, rotations, amplifications and the mixture of these parameters.

Reviewer think that inkjet-printing technique is for mass production and expansion of ink materials. However, unclonable "flower-like" dot patterns is not suitable for cryptography. The latest security technology, the physically unclonable function (PUF) has its own private key and these values should never be replicated. But the author is described and demonstrated the "flower-like" dot patterns shapes that cannot be duplicated. It does not have any information inside "flower-like" dot patterns as an identifier. In addition, the decoding method using deep learning is just a classification of the degree of similarity to the given pictures.

Overall, this paper is not considered to be a new anti-counterfeit technique.

Reviewer #3 (Remarks to the Author):

This manuscript presents a method for creating macroscopic security marks which incorporate unique stochastic patterns within the individual ink droplets. The formation of the patterns in the drying droplets is induced using a preparatory layer of PMMA particles on the substrate. The authors refer to these patterns as physical unclonable functions (PUF). The authors suggest a security system in which these PUFs are characterized and stored in a database by the manufacturer. The end user then confirms the authenticity of the product using a Smartphone.

This is an intriguing idea which could potentially represent a significant advance over previous work in this area. There are, however, several issues that should be addressed prior to publication.

First, since this is an applications paper, the authors should emphasize the advantages the current system would have over PUF's based on the intrinsic surface topography of the material itself (scratch patterns, fiber weave, etc.).

The authors appear to provide adequate information for other groups to reproduce the image creation. Even if other workers produce somewhat different PMMA pre-print surfaces, as long as pinning occurs, there is a good chance that the unique stochastic patterns will be produced during the drying process. I do not, however, feel that enough detail has been provided to understand how the image data base would be created and how the encoded patterns would be read. The authors state that 500 images of the printed patterns were acquired for the initial characterization. Was every droplet in the image recorded? More detail is required regarding the procedure for collecting these 500 images.

Much more functional detail is required also as to how the pattern is read with a Smartphone. How is the necessary magnification and resolution achieved? The area being imaged on the Smartphone in Figure S7 must be less than 1 mm across. Does it matter which section of the image is recorded by the Smartphone, or is the entire image captured? If there is an area within the image that must be captured, then how is this area located by the end user?

The authors state that the patterns are coated with a sticky gel. How do fingerprints and other disturbances of the gel coat affect one's ability to image and decode?

"The surface decoration of print substrates, such as glass, plastic or paper, with randomly distributed poly(methyl methacrylate) (PMMA) nanoparticles is critical for the successful inkjet printing of unclonable security labels."

Finally, the authors tout their method as being inexpensive, a claim which appears to be based primarily on materials cost. However, the surface preparation prior to printing involves multiple washings, plasma cleaning, and subsequent spin coating with PMMA particles prior to inkjet printing. Moreover, the characterization of the stochastic patterns was accomplished using 500 fluorescent images! All of that processing sounds expensive to me. The authors should clarify this issue.

Reviewer #1 (Remarks to the Author):

General comment: *This manuscript is the first example I have seen where physical unclonable functions have been used in combination with machine learning to create an optical authentication system. The PUF is created by printing a luminescent ink on a hydrophobic substrate that contains random defects (random ordering) such that the pattern created when the ink dries can be considered a PUF. The only drawback of the presented method is the need to use a fluorescent microscope to see the authentication pattern. The end-user will not have such a microscope and simple authentication using a smart-phone and e.g. a lens is not possible as the ambient light will disturb the reading of the luminescent PUF patterns. I find that the work and the approach is interesting and can be published if the authors address the three major problems I have identified in their work:*

Response: We appreciate the reviewer for acknowledging the novelty and significance of our work. The reviewer also felt that the use of a (research-based) fluorescent microscope (for a simplest system, size: > 50×50×50 cm; cost: > US\$50,000) that the end-user will not have for pattern readout is the only drawback of the presented work. We agree that it would be a major drawback if such a large, expensive and complicated microscope is required for imaging patterns as the previous reports did (for example, Adv. Mater. 2016, 28, 2330–2336; Nat. Nanotechnol. 2016, 11, 559–565; Sci. Adv. 2018; 4: e1701384, etc.). However, in our original manuscript, we demonstrated the feasibility of using an alternative portable mini-microscope (~120 US dollars), composed of a UV chip, a magnification-adjustable objective lens covered with a cylindrical metal shell and a small WiFi box, to readout the authentication patterns. Such a microscope even is much cheaper than a research-based lens (> 2,000 US\$). More importantly, it also doesn't have the ambient light interference problem as a normal research-based lens does, because the cylindrical metal shell surrounding the lens creates a small dark imaging environment by blocking the ambient light.

To make our manuscript clearer, we revised our manuscript by providing the details of the mini-microscope used for the pattern readout as follows: "Such small, affordable, portable microscopes were utilized by consumers to authenticate the inkjet-printed security labels. The magnification-adjustable objective lens of the portable microscope used here is covered with a cylindrical metal shell that creates a small dark imaging environment by blocking the ambient light (Figure S7)." and "For pattern readout with a smartphone

microscope, the portable microscope was linked to a small WiFi box by a USB line, which allows for the smartphone to control the microscope for real-time imaging (see Supplementary Figure S7)." (line 7, page 8; line 14, page 16). We also revised the Supplementary Figure S7 as follows.

Figure S7 Pattern readout with a smartphone microscope: the portable mini-microscope is composed of a UV chip, a 200 \times magnification-adjustable objective lens covered with a cylindrical metal shell and a small WiFi box. The insert is an enlarged view of the light source and the lens.

Specific comment 1: *The authentication system must be tested in a way that can provide an estimate of the encoding capacity (should be $10E20$ or larger), and the rate of false positives. These two numbers, the latter in particular as it dictates the level of security offered by the system.*

Response: We agree that both the encoding capacity and the rate of false positives are important for an authentication system. Regarding to the encoding capacity issue raised by the reviewer, a universal binary-bit model established by Carro-Temboury *et al.* (Sci. Adv. 2018; 4: e1701384) is adopted for the estimation of the encoding capacity of our security labels. A rectangular coordinate system (typically, the lateral direction is defined as x axis) is required to define the position of each binary-bit unit. However, any rotation of the security label results in a completely different code (Fig. S28; Sci. Adv. 2018; 4: e1701384).

In other word, a real security label will be recognized as a fake one if the end-user rotates or zooms in/out it during the pattern readout process. Alternatively, a mark on the security label is used to define xy axis, which will rotate as the security label rotates (Adv. Mater. 2016, 28, 2330–2336). This guarantees only one code for each security label.

In this work, although the security labels don't contain any marks to define the xy axis, our authentication system allows the end-user to readout the patterns with different image sharpness, brightness, rotations, amplifications and the mixture of these parameters. This is because AI can define the xy axis based on the characteristic features of the security labels (note that we don't know how exactly it did it), which is the beauty of the authentication system described in this work.

We added the calculation process of encoding capacity in **Supplementary Figure S16 and Note S2** as follows. To simplify the encoding capacity calculation, we assume that AI defines the xy axis as shown in Figure S16. We divide the pattern into 30 x 30 arrays, in which a unit is further divided into 5 x 5 arrays of subunits (i.e., Length of each unit, L = 5; Resolution, R = 150). As long as a color (e.g. red) appears in a square subunit, the subunit is labeled as 1; otherwise labeled as 0. A square image of a red flower-like pattern with 750 x 750 pixels was send to AI for the demonstration of our authentication system. R is determined by the lateral pixels of the image and will be even larger than 150. Based on the all flower-like pattern shown in Figure 3a, the pattern filling density (D) in 30 x 30 arrays is in the range of 0.1-0.5. We use D value of 9/25 as an example to estimate the encoding capacity.

According to the binary-bit model established by Carro-Temboury *et al.* (Sci. Adv. 2018; 4: e1701384), the encoding capacity of a flower-like pattern (#codes_F) can be expressed as follows:

$$\#codes_F = \left[C \left(1 + L \left(\frac{1}{\sqrt{D}} - 1 \right) \right)^2 + 1 \right]^{D \frac{R^2}{L^2}} \quad (1)$$

where C is the number of colors of a flower-liker pattern.

Based on eqn 1, the encoding capacity of a red flower-like pattern as shown in Figure S16 is estimated to be $4.7 \times 10^{202} \gg 10^{20}$. For a security label composed of 1,000 red flower-like patterns, its encoding capacity is $(4.7 \times 10^{202})^{1000} > 10^{202,000}$.

Figure S16. An example of the encoding capacity estimation of a red flower-like pattern: $R = 150$, $L = 5$, and $D = 9/25$.

Regarding to the rate of false positives, we authenticate the clear photos from 100 security labels as samples and offer the statistical results that reflect the relationship between the threshold values and the rate of false positives in **Supplementary Table S1**. According the results, when the threshold value is set as 0.4, about 2% security labels are authenticated falsely; if the threshold value is set ≥ 0.5 , the rate of false positives is 0%. Choosing 0.5 as the threshold value in this paper, is in order to correctly authenticate the captured images (from consumers) with different image sharpness, brightness, rotations,

amplifications and the mixture of these parameters, which is the outstanding advantage of our work.

Table S1. The statistical results about the relationship between the threshold values and the rate of false positives for AI authentication.

Threshold values	0.4	0.5	0.6	0.7	0.8	0.9
Rate of false positives	2%	0%	0%	0%	0%	0%

In addition, we added the description as follows “By simply comparing the accuracy on the test with the threshold, the deep learning machine can immediately provide the authentication outcomes (real: accuracy ≥ 0.5 , fake: accuracy < 0.5) to the customers. Regarding to the rate of false positives, we achieved the false positives rate of 0 using the match score of 0.5 as the threshold when sampling 100 security labels (see Supplementary table S1).”

Specific comment 2: *The registration and validation methodology is not reported at a sufficient level of detail. For instance: How are the images indexed, and how is this index read and reported to the matching software? A search of all images to find a match and then report a positive results is probably not how this is done? Are the training images stored? If they are, what is the storage requirement for an individual PUF tag? How fast is the learning protocol? How many spots are used from each label and how are they selected? A fully descriptive walk-through of the procedure is needed.*

Response: We provide the registration and validation methodology with great details in Methods of the revised manuscript. The following is the information for the question raised by the referee.

Q1: *How are the images indexed, and how is this index read and reported to the matching software? A search of all images to find a match and then report a positive results is probably not how this is done?*

Response: We added the description in **Methods** as follows: “We created a file named as gn (n= 1, 2, 3, ...) per security label to store the corresponding 500 training images prior to the training process. The training images stored in the file gn are named as gn_000,

gn_001, ..., gn_500. Many files from these security labels composed a database. After the 500 training images of a security label were learnt by AI, their structural information was remembered and linked to the file name gn (e.g., g1). Then the training images will be deleted. When consumers randomly take a picture of a real security label and sent it to the AI, the AI can automatically recall the accurately corresponding relationship and output the indexing name with a detailed match score. According our results, if the captured image from the end-user is clear enough, the match score of the image of true security label is more than 99% (Figure 4e). On the other hand, if the image (from a fake label) has never been learned, the engine will give a lower match score (Figure 4f).” (paragraph 3, page 17).

Q2: Are the training images stored? If they are, what is the storage requirement for an individual PUF tag?

Response: The training images don't need to be stored and are not stored in this case. After training, the images for training can be deleted. Notably, the storage requirement is mainly determined by the neural network itself, about 200 M Bytes here. That is to say, the storage requirement does not increase with the number of samples because the images for training are deleted after leaning. We added the description in **Methods** as follows: “The training images don't need to be stored and are not stored in this case; The storage requirement is mainly determined by the neural network itself, about 200 M Bytes here.” (line 12, page 17)

Q3: How fast is the learning protocol?

Response: We added the description in **Methods** as follows: “For the AI technology we used, the learning process takes 2 hours in our case.” (line 14, page17). This time can be greatly shortened by using a more advanced neural network.

Q4: How many spots are used from each label and how are they selected?

Response: The number of spots from each label and their selection can be customized. Ideally, three adjacent spots of flower-like patterns from a security label are preferred for authentication as each photo captured by a smartphone microscope at 200x magnification contains three dots as shown in Figure S7. In this work, as proof-of-concept, only single spot was printed and used to demonstrate the authentication procedure (Figure 4).

Specific comment 3: *The authors have not cited the landmark reports in the field. There are several important papers that are not mentioned, these must be included, these 5*

papers are listed below. There are several other papers on PUFs that could also be included, these are also listed below.

These references needs to be read and cited:

Prior to publication these references must be included as they are the first examples of the use of PUFs for authentication.

- Horstmeyer, R.; Judkewitz, B.; Vellekoop, I. M.; Assawaworrarit, S.; Yang, C., Physical key-protected one-time pad. Sci. Rep. 2013, 3, 3543.

- Takahashi, T.; Kudo, Y.; Ishiyama, R. In Mass-produced Parts Traceability System Based on Automated Scanning of “Fingerprint of Things”, Fifteenth IAPR International Conference on Machine Vision Applications (MVA), Nagoya University, Nagoya, Japan, Nagoya University, Nagoya, Japan, 2017.

- Carro-Temboury, M. R.; Arppe, R.; Vosch, T.; Sørensen, T. J., An optical authentication system based on imaging of excitation-selected lanthanide luminescence. Science Advances 2018, 4 (1), e1701384.

- Wigger, B.; Meissner, T.; Forste, A.; Jetter, V.; Zimmermann, A., Using unique surface patterns of injection moulded plastic components as an image based Physical Unclonable Function for secure component identification. Scientific reports 2018, 8 (1), 4738.

The authors are strongly encourages to also read and include these refences:

- Bae, H. J.; Bae, S.; Paul, C.; Han, S.; Kim, J.; Kim, L. N.; Kim, K.; Song, S.-H.; Park, W.; Kwon, S., Biomimetic microfingerprints for anti-counterfeiting strategies. Adv. Mater. 2015, 27 (12), 2083-2089.

- Smith, A. F.; Patton, P.; Skrabalak, S. E., Plasmonic nanoparticles as a physically unclonable function for responsive anti-counterfeit nanofingerprints. Adv. Funct. Mater. 2016, 26 (9), 1315-1321.

- Geng, Y.; Noh, J.; Drevensek-Olenik, I.; Rupp, R.; Lenzini, G.; Lagerwall, J. P. F., High-fidelity spherical cholesteric liquid crystal Bragg reflectors generating unclonable patterns for secure authentication. Sci. Rep. 2016, 6, 26840.

- Hu, Z.; Comeras, J. M. M. L.; Park, H.; Tang, J.; Afzali, A.; Tulevski, G. S.; Hannon, J. B.; Liehr, M.; Han, S.-J., Physically unclonable cryptographic primitives using self-assembled

carbon nanotubes. *Nature Nanotechnol.* 2016, 11 (6), 559-565.

- Kim, J.; Yun, J. M.; Jung, J.; Song, H.; Kim, J.-B.; Ihee, H., *Anti-counterfeit nanoscale fingerprints based on randomly distributed nanowires.* *Nanotechnology* 2014, 25 (15), 155303.

- Tian, L.; Liu, K.-K.; Fei, M.; Tadepalli, S.; Cao, S.; Geldmeier, J. A.; Tsukruk, V. V.; Singamaneni, S., *Plasmonic nanogels for unclonable optical tagging.* *ACS Appl. Mater. Interfaces* 2016, 8 (6), 4031-4041.

- Herder, C.; Yu, M. D.; Koushanfar, F.; Devadas, S., *Physical unclonable functions and applications: A tutorial.* *Proc. IEEE* 2014, 102 (8), 1126-1141.

Response: All the references mentioned by the reviewer have been included in the revised manuscript.

Specific comment 4: Please also correct the following minor issues.

(1) *The spots in figure 3 and 4 are clearly different in shape and form, please comment.*

Response: We agree with referee that the spots in Figure 3 and 4 looks different in shape and form. As discussed in the original manuscript, the shape of each spot relies on the PMMA layer with randomly distributed nanoparticles and the ink evaporation rate (sensitively affected by the temperature and humidity of the ambient environment), which are not replicable for each printing.

(2) *Line 20: The method is not smartphone readable, must be deleted. It is read using a fluorescent microscope attachment connected to a smartphone*

Response: We have deleted the “smartphone readable” accordingly.

(3) *Line 21: nothing is unlimited, a number must be reported, see also 1) above.*

Response: We agree that the encoding capacity is not infinite. We deleted the statement of “with unlimited variability of 2D patterns” of the sentence mentioned by the reviewer (line 23, page 1). A number of the encoding capacity of the security labels calculated based on Carro-Temboury’s model (*Sci. Adv.* 2018; 4: e1701384) is provided in the revised

manuscript as follows: “According to Carro-Temboury’s model, the encoding capacity of a red flower-like PUF pattern, I , is calculated to be 4.7×10^{202} (see Supplementary S16 for calculation details). Therefore, for a security label composed of 1,000 red flower-like PUF patterns, its encoding capacity will be larger than $10^{202,000}$.” (line 12, page 10).

(4) Line 30: not read using a smartphone, please delete.

Response: We have deleted the “smartphone” in Line 30.

(5) Line 30: for the AI technique please comment on the demand of computer power and storage space

Response: We added the details of the computer used in **Methods** as follows “the CPU is Intel(R) Core(TM) i7-6700 CPU @3040GHz; the GPU is NVIDIA GTX 1080; the RAM is 32.0 GB; HDD Capability is 1 TB; the computer rated power is 350 W/h”(line 15, page 17). It is worthwhile to mention that this is a normal desktop that can run the neural work (i.e., deep learning neural network, about 200 M Byte). No special requirements for the computer are needed. We added the details of the computer used in **Methods** as follows: “The training images don’t need to be stored and are not stored in this case; The storage requirement is mainly determined by the neural network itself, about 200 M Bytes here.” (line 12, page 17)

(6) Line 56: Sentence must be referenced, use e.g. Arppe et al.

Response: The sentence is referenced with the following papers: “Science Advances 2018, 4 (1), e1701384” and “Nature Nanotechnol. 2016, 11 (6), 559-565.”

(7) Line 57: Sentence must be referenced, use e.g. Arppe et al.

Response: The sentence is referenced with the following paper: “Science Advances 2018, 4 (1), e1701384.”

(8) Line 62: Sentence must be referenced, I am not aware of a reference that can support the claim.

Response: The sentence is referenced with the following papers: “Adv. Optical Mater. 2016, 4, 1915–1932”, and “Nanoscale, 2017, 9, 15982–15989”.

(9) Line 66: Naked eye authentication is not mentioned below, please delete.

Response: Thank you for your comment. However, we believe that “Naked eye authentication” is important in the original manuscript. The original description is that “Furthermore, the inkjet-printed macroscale security labels allow fast, frequent authentication by the naked eyes or using a smart phone.” We mean that QR codes and bar codes in Figure 2 (about 4 cm in horizontal and vertical directions) composed of thousands of dot patterns can be directly seen by naked eyes under UV irradiation. We provide the corresponding scale bars to the images in Figure 2 and a sentence about naked eye authentication discussion in the revised manuscript as follows: “The property that these images are only seen upon UV excitation offers the first layer of security realized by naked eye authentication of macroscopic patterns; and the PUF nature of the flower-like patterns is the second more secure layer.” (line 10, page 9).

(10) Line 69: Sentence must be referenced.

Response: The sentence is referenced with the following papers: “Science 2002, 297, 2026-2030”

(11) Line 70: Sentence need more references, use e.g. Arppe et al., Carro-Temboury et al. and Takahashi et al.

Response: The sentence is referenced with the following papers: “Nature Mater 2017, 1, 0031”, “Science Advances 2018, 4 (1), e1701384.” and “Fifteenth IAPR International Conference on Machine Vision Applications (MVA), Nagoya University, Nagoya, Japan, Nagoya University, Nagoya, Japan, 2017.”

(12) Line 82: Sentence must be referenced.

Response: The sentence is referenced with the following paper: “Adv Mater 2015, 27, 2083-2089”.

(13) Line 96: *This is a postulate that needs to be substantiated*

Response: We provide the discussion in the revised manuscript as follows: "According to Carro-Temboury's model, the encoding capacity of a red flower-like PUF pattern, I, is calculated to be 4.7×10^{202} (see Supplementary S16 for calculation details). Therefore, for a security label composed of 1,000 red flower-like PUF patterns, its encoding capacity will be larger than $10^{202,000}$." (line 12, page 10) (see Specific Comment 1 for more details).

(14) Line 99: *the claim that they are quite different needs to be documented and reported as a number e.g. rate of false positives or encoding capacity.*

Response: Thanks for your comments. the original description of "however, at the microscopic level, they are quite different from each other." can be documented by the Figure 3, and the corresponding discussion of "By carefully comparing all the pixels within the letters (i.e., the same sample), no identical "flower-like" micropatterns were found. A copy of the red, green and blue fluorescence counterparts of the "FZU" letters shown in Figure 3 fabricated under the same conditions also shows entirely different geometries of the corresponding pixels (see Supplementary Figure S15)."

In addition, we have added the results on the rate of false positives (see Supplementary Table S1) and *encoding capacity* (see Supplementary Note S2), and the corresponding description in revised manuscript. See Response to Specific Comment 1 and 4-(38) for more details.

(15) Line 105: *The methods not convenient for authentication. First it needs a microscope. And second, no time for authentication is given and no detailed method of registration and validation is reported.*

Response: We believe that our authentication methods are convenient for the following reasons. Firstly, our security labels can be verified by naked eye under UV irradiation like fluorescence tag labeled banknotes. Secondly, our PUF pattern authentication method is more convenient than those reported previously in terms of pattern readout tools (cheap portable min-microscopes instead of expensive research-based microscopes or other

equipment), and authentication technique (AI instead of conventional algorithm). The following table listed the main differences between our authentication methods and the previously reported works (especially the works mentioned by the referee). The detailed method of registration and validation including the time for authentication is provided in **Methods** (see Response to Specific comment 2).

No.	Unclonable property	Readout tools/light source	Authentication method	Ref
1	Yes	Research-based microscope/ 532 nm laser	Conventional	Sci. Rep. 2013, 3, 3543
2	Yes	Research-based microscope/ 465, 488, and 450 nm laser	Conventional	Sci. Adv. 2018, 4, e1701384
3	Yes	Industrial camera/ambient light	Conventional	Sci. Rep. 2018, 8, 4738
4	Yes	Research-based microscope/ Hg–Xe lamp	Conventional	Adv. Mater. 2015, 27 2083-2089
5	Yes	Research-based microscope/ halogen lamp	Conventional	Adv. Funct. Mater. 2016, 26, 1315– 1321
6	Yes	TFT current test machine	Conventional	Nature Nanotechnol. 2016, 11, 559.
7	No	Naked eye	Conventional	Sci. Rep. 2016, 6, 30885
8	Yes	Research-based microscope/ 785 nm laser	Conventional	ACS Appl Mater Inter, 2016, 8(6): 4031-4041.
9	Yes	Portable mini-microscope/LED	AI	This work

(16) Line 112: *Oil-phase route*, is this the right word?

Response: We revised the phrase “These quantum dots were synthesized using a well-established oil-phase route” by “These quantum dots were synthesized using a well-established chemical route” (line 16, page 5).

(17) Line 118: I cannot decipher this sentence, please rephrase

Response: We revised the description as follows: "Generally, for quantum dots, the relationship between size and emissive wavelength accords with quantum confinement effect — that a small size corresponds to a larger band gap and emits short-wavelength fluorescence.²⁷ In this case, the emission wavelength of red, green and blue quantum dots shown in Figure 1a was determined by the composition rather than the size of quantum dots (see Supplementary Figure S1c).²⁵" (line 22, page 5).

(18) Line 120: sentence/claim must be referenced

Response: The sentence is referenced with the following paper: Chem Mater 2017, 29, 3644-3652.

(19) Line 122: sentence/claim must be referenced

Response: The sentence is referenced with the following paper: Nature Photonics 2015, 9, 259-266.

(20) Line 133: Please document that the method works on different substrates by including images in the SI, otherwise delete.

Response: We have deleted the description accordingly.

(21) Line 134 various solvents, either specify or refer to methods section

Response: We refer various solvents used for sonication cleaning to Methods section of the revised manuscript as follows: "Prior to inkjet printing, the print indium-tin-oxide (ITO)-coated glasses were cleaned by sonication in various solvents (see Methods for detailed sonication cleaning procedure)" (line 11, page 6). We offered the detailed description of various solvents in **Methods section** "Indium-tin-oxide (ITO)-coated glass substrates were cleaned with ultrasonication successively in deionized (DI) water, acetone, isopropanol and DI water." (line 17, page 15).

(22) Lines 140-141: undocumented claim, please document.

Response: We revised the discussion as: “Such randomly arranged pinning points are critical for the successful inkjet printing of unclonable security labels.” (line 18, page 6).

In addition, this sentence is documented with the results in Supplementary Figure S6. For same substrate without PMMA modification, similar and repeatable patterns are formed. In addition, for same substrate with PMMA modification without pinning points, uniform and repeatable patterns are formed as well. However, for the substrate with PMMA with pinning points, unclonable flower-like patterns appear (Figure 1f). In conclusion, randomly arranged pinning points are critical for the successful inkjet printing of unclonable security labels.

(23) Lines 149-152: The time of printing and drying is critical to mass-production. How fast is the total process?

Response: The printing time is mainly determined by the substrate moving time and the number of lines intended to print. We used flying mode for inkjet printing: the substrate moving time for each line which includes 2000 points is less than 0.5 s. During our process, we can finish 800 line in 5 min, corresponding 1600,000 unclonable patterns after drying. If every single macroscopic pattern has 1000 unclonable patterns, we can obtain 1600 macroscopic patterns in 5 min. Furthermore, the time of process can also be dramatically cut down by using commercialized machine with 128 printing nozzles.

The time of drying process is about 5 min for each batch of security labels. In addition, the printing and drying process can also be performed simultaneously for mass production.

In light of referee’s comment, we added the following statement in **Methods** as follows: “If every single macroscopic pattern has 1000 unclonable flower-like points, we can achieve 1600 macroscopic patterns in 5 min with our single-nozzle printing machine. The time of drying is about 5 min for each batch of security label” (line 25, page 15).

(24) Line 157: I cannot decipher this sentence, please rephrase

Response: We rephrase the sentence by “With the shrinking of droplet, the volume-smaller droplet is more liable to be tortured by pinning points, thus splitting into several smaller sub-droplets (see Supplementary Figure S5).” (line 11, page 7).

In addition, this torturing and splitting process of such volume-smaller droplet was recorded by monitoring the shape evolution of a droplet as a function of time (see Supplementary Figure S5e-l)

(25) Line 165: Sticky-gel film? Does not make sense, please give details and exact procedure to laminate codes in Methods section.

Response: The sticky-gel film is used to protect the security labels during the circulation (i.e., for real application purpose), just like the PVP used in Sci. Adv. 2018;4: e1701384. We add the detailed procedure to laminate codes in **Methods** as follows: "Sticky-gel film with gel thickness of 1.5 mm and retention level of X4 was purchased from Gel-Pak company." and "The as-fabricated security labels were covered with gel films by tearing of their polycarbonate coversheet and then stick their gel material on the labels for stability test (see Supplementary Figure S17)." (line 11, page 14; line 1, page 16).

;

(26) Line 179-180: Please mention the issue of have consumers use this type of equipment, and be critical as it is a major issue.

Response: Thanks for your comments. We have added the description as follows: "Such small, affordable, portable microscopes were utilized by consumers to authenticate the inkjet-printed security labels." (line 7, page 8). And the images of the portable microscope used here are shown in Figure S7.

We demonstrated the feasibility of using an alternative portable mini-microscope (~120 US\$), composed of a UV chip, a magnification-adjustable objective lens covered with a cylindrical metal shell and a small WiFi box, to readout the authentication patterns (see line 14, page 16 and Supplementary Figure S7). Such a microscope is much cheaper than a research-based fluorescent microscope (for a simplest system, size: > 50×50×50 cm; cost: > US\$50,000). For more details, see Response to Specific Comment 4-(15).

(27) Lines 188-189: Please supply all data for QY determination as SI

Response: We add the experimental details for QY determination in **Methods** of the revised manuscript as follows (paragraph 3, page 16).

Quantum yield (QY) of quantum dots: The results are obtained by comparing integrated PL intensities using the standard procedure. The QYs of blue, green and red emission quantum dots were measured relative to Coumarin 480 (QY 99% in ethanol) with excitation at 350 nm, Coumarin 480 (QY 99% in ethanol) with excitation at 370 nm and rhodamine 6G (QY 95% in ethanol) with excitation at 450 nm, respectively. Solutions of quantum dots in toluene were optically matched at the excitation wavelength. Fluorescence spectra of quantum dots and dye were taken under identical spectrometer conditions in triplicate and averaged. The optical density was kept below 0.06 at the λ_{\max} , and the integrated intensities of the emission spectra, corrected for differences in index of refraction and concentration, were used to calculate the quantum yields using the expression.

$$\text{QY of quantum dots} = \text{QY}_R \times \frac{I}{I_R} \times \frac{A_R}{A} \times \frac{n^2}{n_R^2}$$

where QY is the quantum yield, I is the measured integrated PL emission intensity, n is refractive index ($n = 1.496$ for toluene; $n = 1.361$ for ethanol) and A is the optical density at the excitation wavelength.

In addition, we have now added all the original data for calculating the QY of blue quantum dots as an example in Supplementary Note S2.

(28) Lines 203-207: Please make sure to state that all these images are only seen upon UV excitation. It is a good feature as that is the first layer of security, the PUF nature is the second more secure layer.

Response: Thanks for the valuable suggestion. We provide a sentence about naked eye authentication discussion in the revised manuscript as follows: "The property that these images are only seen upon UV excitation offers the first layer of security realized by naked eye authentication of macroscopic patterns; and the PUF nature of the flower-like patterns is the second more secure layer." (line 10, page 9).

(29) Line 211: Please cite the lanthanide complexes used in euro bank notes. Andres et al Adv. Mater.

Response: The sentence is referenced with the following paper: "Andres J, Hersch RD,

Moser JE, Chauvin AS. A New Anti-Counterfeiting Feature Relying on Invisible Luminescent Full Color Images Printed with Lanthanide-Based Inks”, (Adv. Funct. Mater. 2014, 24, 5029-5036).

(30) Lines 214-215: *The procedure described here cannot be mass produced. Please comment.*

Response: Although the procedure involves multiple washings, plasma cleaning, and spin coating, some previous reports have shown that it can be used for mass production of TV screens (Adv. Mater. 2009, 21, 2151–2155; Sci. China Chem. 2017, 60,10 and ACS Appl. Mater. Interfaces 2016, 8, 26162-26168). More impressively, top companies in display like BOE company, TCL company and JOLED company have established similar procedures for display production and model display products have been achieved (<http://en.silkroad.news.cn/2018/1128/121822.shtml>; <https://www.oled-info.com/johua-printing-developed-ink-jet-printed-31-4k-oled-panel>; <http://olednet.com/joled-oled-worlds-first-inkjet-printing/>). Therefore, we believe the procedure here can be used for mass production of security labels.

(31) Line 231: The encoding capacity is not infinite. Please consult the detailed considerations in the supporting information of Carro-Temboury et al.

Response: We agree that the encoding capacity is not infinite. We have estimated the encoding capacity according to the method in the supporting information of Carro-Temboury et al. we added the evaluation of encoding capacity in revised manuscript as follows: “According to Carro-Temboury’s model, the encoding capacity of a red flower-like PUF pattern, I, is calculated to be 4.7×10^{202} (see Supplementary S16 for calculation details). Therefore, for a security label composed of 1,000 red flower-like PUF patterns, its encoding capacity will be larger than $10^{202,000}$.” (line 12, page 10). See Response to Specific Comment 1 for more details.

(32) Line 240: *Sentence must be referenced.*

Response: The sentence is referenced with the following papers: “Adv Mater 2015, 27,

2083-2089” and Adv Mater 2016, 28, 2330-2336”

(33) Line 252: *The number and types of characteristic features must be given and described.*

Response: Thanks for your comment. our flower-like patterns do have characteristic features including locations, colors and edges. And we provide the encoding capacity in revised manuscript (see Specific Comment 1 for more details). Unfortunately, we are not able to provide the number and the types of characteristic features that the deep learning engine learns at this moment. This is because we and even the designers of deep learning engine do not know exactly what characteristic features it learns and how it learns specifically. It will be a long way to understand how the deep learning learns, just like trying to understand how a human learns. Despite of this, we can still take advantage of deep learning technology to do image analysis and comparison as described in this work, even plan chemical syntheses (Nature 2018, 555, 604–610) and analyze TEM images (ACS Nano 2017, 11, 12742–12752). In addition, the AlphaGo based on deep learning from Google have beaten top Go Masters in 2016. People still cannot clarify the number and types of characteristic features that AlphaGo learns from humans. Hopefully, an evolution process of a simple neural network in the following link can help the referee to understand the learning process (<https://playground.tensorflow.org>).

(34) Line 256: *IT is not just a smartphone, delete or rephrase.*

Response: We replace “smartphones” with “portable mini-microscope-connected smartphones”. (line 16, page 11).

(35) Line 258: *How long does authentication take, how is it done, and how many can be run in parallel? This is critical.*

Response: We add the description in **Methods** of the revised manuscript as follows: “The authentication process takes about 2 seconds.” The process about how the authentication is done has been added in **Methods** as follows: “*Registration and validation methodology:* We created a file named as gn (n= 1, 2, 3, ...) per security label to store the corresponding

500 training images prior to the training process. The training images stored in the file gn are named as gn_000, gn_001, ..., gn_500. Many files from these security labels composed a database. After the 500 training images of a security label were learnt by AI, their structural information was remembered and linked to the file name gn (e.g., g1). Then the training images will be deleted. When consumers randomly take a picture of a real security label and sent it to the AI, the AI can automatically recall the accurately corresponding relationship and output the indexing name with a detailed match score. According our results, if the captured image from the end-user is clear enough, the match score of the image of true security label is more than 99% (Figure 4e). On the other hand, if the image (from a fake label) has never been learned, the engine will give a lower match score (Figure 4f).” (paragraph 3, page 17)

Each end-user has an app in mobile phone as the input and output port to AI, which is independent and does not interfere with each other. In principle, it allows "innumerable" individuals using the technique at the same time from computer science and Internet science point of view.

(36) Line 264: Which exact same geometrical characteristics?

Response: During the training process, a clear image of a flower-like pattern was rotated by a step of 0.72° for 360° using an algorithm, producing 500 learning images at each step. These images are sent to AI to learn the unique geometrical characteristics of the images. Therefore, the geometrical characteristics of the rotated images are exact same as the un-rotated one, as they are from the same image.

(37) Line 268: A number close to 1 can be many things, please be more concrete.

Response: We rephrase the description with “After about 1000 learning cycles, they can be recognized with a match score between 97% to 100% when being sent to AI for validation again (see Supplementary Figure S19)” (line2, page 12).

(38) Lines 276-283: Please use the actual match scores, the threshold values and the non-match scores to calculate the rate of false positives and the actual encoding capacity.

The latter must be a function of the threshold value.

Response: We cannot agree more that the threshold values will affect the rate of false positives and the actual encoding capacity.

To evaluate the rate of false positives, we authenticate the clear photos from 100 security labels as samples and offer the statistical results that reflect the relationship between the threshold values and the rate of false positives in **Supplementary Table S1**. According the results, when the threshold value is set as 0.4, about 2% security labels are authenticated falsely; if the threshold value is set ≥ 0.5 , the rate of false positives is 0%. Choosing 0.5 as the threshold value in this paper, is in order to correctly authenticate the captured images (from consumers) that are with different image sharpness, brightness, rotations, amplifications and the mixture of these parameters, which is the outstanding advantage of our work.

Table S1. The statistical results about the relationship between the threshold values and the rate of false positives for AI authentication.

Threshold values	0.4	0.5	0.6	0.7	0.8	0.9
Rate of false positives	2%	0%	0%	0%	0%	0%

Since we don't know very exactly how the deep learning engine learns, built the links and remember the links between image and corresponding file name (e.g., g1), we cannot calculate the actual encoding capacity based on the threshold values specifically. Although the actual encoding capacity may be lower than the value estimated by Carro-Temboury's model (See Response to Specific Comment 1), it must be much larger than 10^{20} (a minimum value requested for real application). And if choosing 0.5 as the threshold value, we can obtain the actual match scores of 100%.

We added the description as follows "By simply comparing the accuracy on the test with the threshold, the deep learning machine can immediately provide the authentication outcomes (real: accuracy ≥ 0.5 , fake: accuracy < 0.5) to the customers. Regarding to the rate of false positives, we achieved the false positives rate of 0 using the match score of 0.5 as the

threshold when sampling 100 security labels (see Supplementary table S1).” (line 18, page 12).

(39) Line 286: Time claim must be validated.

Response: We add the description in **Methods** of the revised manuscript as follows: “The authentication process takes about 2 seconds.” (line 4, page 18).

(40) Line 300: Again. A smartphone is not needed, you need a microscope.

Response: We delete “A smartphone” accordingly and revise the relevant discussion as follows: “A more reliable authentication strategy by using AI techniques has been developed.” (line 12, page 13).

(41) Line 305: This is not true. There is not an easy authentication without the use of specialized equipment and the printing method/drying time is not compatible with modern means of mass production. And neither is ITO and spin-casting. It is a good step closer to a compatible method, but we are not there yet.

Response: We agree that the anti-counterfeiting technology and the printing method developed here is a good step closer to a compatible method. Therefore, we rephrase the description with “The anti-counterfeiting technology described in this work is a good step closer to commercial applications” (line 17, page 13).

(42) Lines 324-: Is stirring used at all?

Response: Yes, stirring is needed throughout the synthesis of core-shell quantum dots.

(43) Line 378: the symbol before 500 is missing

Response: We delete the typo (i.e. the symbol) before 500.

(44) Line 310: Again. A smartphone is not needed, you need a microscope.

Response: See Response of Specific Comment 4-(34). We rephrase “A smartphone” with “a portable mini microscope”. (line 25, page 13).

Reviewer #2 (Remarks to the Author):

General comment: *Author reported a non-destructive, inkjet-printable, smart-phone readable, AI decodable and unclonable security label. In this process, author addressed 1) before inkjet printing, the print substrate with random-distributed pinning points of PMMA nanoparticles on PMMA film is used. 2) 2D patterned security labels composed of red, green or blue arrays are fabricated through inkjet printing using II-VI semiconductor core-shell quantum dots as inks. 3) forming physically unclonable "flower-like" patterns using the creation of stochastic pinning points at the three-phase contact line of the ink droplets. 4) authentication is done by deep learning decoding mechanism. Five hundred fluorescence images of each security label obtained by randomly shifting and rotating a same image are provided to AI for learning and classifying. The threshold of the accuracy at a value of 0.5 is then set to distinguish the real and fake security labels. For comparison, six fake security labels were sent to AI for authentication then the corresponding accuracy is almost zero for all the fake security labels. Author conclude that the inkjet-printing technique guarantees the mass production of security labels at low cost and the developed authentication strategy allows for the fast authentication of the covert, unclonable "flower-like" dot patterns with different sharpness, brightness, rotations, amplifications and the mixture of these parameters.*

Response: We appreciate referee for the acknowledgment of the advantages of our work in the mass production of security labels at low cost and the developed authentication strategy.

Specific Comment 1: *Reviewer think that inkjet-printing technique is for mass production and expansion of ink materials. However, unclonable "flower-like" dot patterns is not suitable for cryptography. The latest security technology, the physically unclonable function (PUF) has its own private key and these values should never be replicated. But the author is described and demonstrated the "flower-like" dot patterns shapes that cannot be duplicated. It does not have any information inside "flower-like" dot patterns as an identifier.*

Response: We agree with reviewer that the physically unclonable function (PUF) based security labels have their own private keys that are hard to be replicated.

According to the landmark reports in the field, the PUF-based private keys are generally hidden in a random pattern, like randomly distributed nanoparticle patterns (Sci. Adv. 2018;

4: e1701384), artificial fingerprints (Adv. Mater. 2015, 27, 2083–2089), and the intrinsic surface topography of a material (Science 2002, 297, 2026-2030), etc. Like the random features mentioned above, the “flower-like” dot patterns described in this work can be easily transformed to binary-bit private keys. Based on the Carro-Temboury’s model, their coding capacity is estimated as high as 4.7×10^{202} , making the private keys unclonable. In light of the referee’s comment, we added the calculation process of encoding capacity in **Supplementary Figure S16 and Note S2**.

Specific Comment 2: *In addition, the decoding method using deep learning is just a classification of the degree of similarity to the given pictures.*

Response: We agree with reviewer that the decoding method using deep learning is a classification of the degree of similarity to the given pictures, which carry PUF codes. This is the first example where PUFs have been used in combination with deep learning to create an optical authentication system. Compared with the conventional algorithms, the authentication system developed in this work doesn’t need any marks to define the xy axis of a rectangular coordinate system and allows the end-user to readout the patterns with different image sharpness, brightness, rotations, amplifications and the mixture of these parameters.

Reviewer #3 (Remarks to the Author):

General comment: *This manuscript presents a method for creating macroscopic security marks which incorporate unique stochastic patterns within the individual ink droplets. The formation of the patterns in the drying droplets is induced using a preparatory layer of PMMA particles on the substrate. The authors refer to these patterns as physical unclonable functions (PUF). The authors suggest a security system in which these PUFs are characterized and stored in a database by the manufacturer. The end user then confirms the authenticity of the product using a Smartphone.*

This is an intriguing idea which could potentially represent a significant advance over previous work in this area. There are, however, several issues that should be addressed prior to publication.

Response: We appreciate for the valuable and positive comments from the referee.

Specific Comment 1: *First, since this is an applications paper, the authors should emphasize the advantages the current system would have over PUF's based on the intrinsic surface topography of the material itself (scratch patterns, fiber weave, etc.).*

Response: We emphasize the advantages of our system in the revised manuscript as follows: "Compared with previous reports using the intrinsic surface topography of a material (e.g., scratch patterns, fiber weave, etc.) for PUF encoding, the system presented here has many advantages: (1) the developed printing strategy for the security label fabrication not only allows for various pattern design but also makes the mass-production at low cost possible; (2) the quantum dot ink is fluorescence active, guaranteeing the readout signals from suffering the interference by fingerprints and dusts; (3) the quantum dot security labels are only visible upon UV excitation, which offers the first layer of security, while the PUF nature originated from the random flower-like dot patterns provides the second more secure layer; (4) the first layer of security can be easily authenticated with naked eyes, while the second layer of security is able to be authenticated with AI technique rather than time-consuming machine learning algorithms."(line 21, page 4)

Specific Comment 2: *The authors appear to provide adequate information for other groups to reproduce the image creation. Even if other workers produce somewhat different*

PMMA pre-print surfaces, as long as pinning occurs, there is a good chance that the unique stochastic patterns will be produced during the drying process. I do not, however, feel that enough detail has been provided to understand how the image database would be created and how the encoded patterns would be read. The authors state that 500 images of the printed patterns were acquired for the initial characterization. Was every droplet in the image recorded? More detail is required regarding the procedure for collecting these 500 images.

Response: We agree with the reviewer that, like all other PUF-based security labels, there still is a chance to produce the unique stochastic patterns presented in this work. However, the chance to reproduce them is very low (the probability is almost zero) due to the high encoding capacity of the patterns (see **Supplementary Figure S16 and Note S2**). For example, a recent work reported by Carro Temboury et al. showed an extremely low probability of 10^{-100} to produce a ~80%-match nanoparticle distribution pattern composed of 10,000 pixels (see Supplementary Fig. 24d, Sci. Adv. 2018; 4: e1701384). Since a “flower-like” pattern developed here contains pixels much higher than 10,000, the probability to duplicate it will be very low.

To demonstrate the proof-of-concept, only one dot from the lower-left of every bar code is used for illustrating the decoding mechanism as is discussed in Figure 4. In reality, the first three dots of a complex security label will be used for authentication as each photo captured by a smartphone microscope as shown in Figure S7.

We added the description of collecting 500 training images in Methods as follows: “For a typical process, one lower-left dot representing a security label is captured as an image. Such a clear image was rotated by a step of 0.72° for 360° using an algorithm, producing a set of 500 training images.” (line 8, page 17).

We added detailed registration and validation methodology as follows: “*Registration and validation methodology:* We created a file named as gn (n= 1, 2, 3, ...) per security label to store the corresponding 500 training images prior to the training process. The training images stored in the file gn are named as gn_000, gn_001, ..., gn_500. Many files from these security labels composed a database. After the 500 training images of a security label were learnt by AI, their structural information was remembered and linked to the file name gn (e.g., g1). Then the training images will be deleted. When consumers randomly take a picture of a real security label and sent it to the AI, the AI can automatically recall the accurately corresponding relationship and output the indexing name with a detailed match score. According our results, if the captured image from the end-user is clear enough, the

match score of the image of true security label is more than 99% (Figure 4e). On the other hand, if the image (from a fake label) has never been learned, the engine will give a lower match score (Figure 4f)." (line 19, page 17).

Specific Comment 3: *Much more functional detail is required also as to how the pattern is read with a Smartphone. How is the necessary magnification and resolution achieved? The area being imaged on the Smartphone in Figure S7 must be less than 1 mm across. Does it matter which section of the image is recorded by the Smartphone, or is the entire image captured? If there is an area within the image that must be captured, then how is this area located by the end user?*

Response: Thank you very much for your important and detailed comments. In fact, the decoding process is achieved by a portable microscope connecting with a mobile phone by Wireless Fidelity. The commercially available portable microscope has a necessary magnification of 200x, and four UV chips (see Supplementary Figure S7). Such information is provided in the revised manuscript (line 8, page 7).

In light of the referee's feedback, we have now corrected all the according descriptions, added the enlarged view of the portable microscope in Supplementary Figure S7, and modified the description as follows: "For pattern readout with a smartphone microscope, the portable microscope was linked to a small WiFi box by a USB line, which allows for the smartphone to control the microscope for real-time imaging (see Supplementary Figure S7)." (line 14, page 16).

Figure S7 Pattern readout with a smartphone microscope: the portable mini-microscope is composed of a UV chip, a 200x magnification-adjustable objective lens covered with a cylindrical metal shell and a small WiFi box. The insert is an enlarged view of the light source and the lens.

For the authentication, we used two-step verification procedures. More specifically, we exposed a macroscopic security label (e.g., FZU logo) to UV light and observed the whole image of the security label (first step). Then we used the portable microscope to locate the desired spot (i.e., the last dot pattern from the lower-left of the security label) for imaging. To demonstrate the proof-of-concept, one dot pattern representing a security label is used for illustrating the decoding mechanism as is discussed in Figure 4.

Specific Comment 4: *The authors state that the patterns are coated with a sticky gel. How do fingerprints and other disturbances of the gel coat affect one's ability to image and decode?*

Response: The dusts and the end-user's fingerprints and dusts adsorbed on the gel do not affect the ability to image and decode our security labels as they are not fluorescent under UV irradiation. The gel coat to protect the security labels from destruction during the circulation like other security labels has been demonstrated (e.g., *Adv. Mater.* 2016, 28, 2330–2336).

Specific Comment 5: *“The surface decoration of print substrates, such as glass, plastic or paper, with randomly distributed poly(methyl methacrylate) (PMMA) nanoparticles is critical for the successful inkjet printing of unclonable security labels.” Finally, the authors tout their method as being inexpensive, a claim which appears to be based primarily on materials cost. However, the surface preparation prior to printing involves multiple washings, plasma cleaning, and subsequent spin coating with PMMA particles prior to inkjet printing. Moreover, the characterization of the stochastic patterns was accomplished using 500 fluorescent images! All of that processing sounds expensive to me. The authors should clarify this issue.*

Response: The surface preparation prior to printing involves multiple washings, plasma cleaning, and subsequent spin coating with PMMA particles prior to inkjet printing has been shown to be cheap procedures and widely used in industry for the production of display panels. Top companies in display like BOE company, TCL company and JOLED company have established similar procedures for display production and model display products have been achieved (<http://en.silkroad.news.cn/2018/1128/121822.shtml>; <https://www.oled-info.com/johua-printing-developed-ink-jet-printed-31-4k-oled-panel>; <http://olednet.com/joled-oled-worlds-first-inkjet-printing/>). Therefore, we believe the procedure can be used for mass production of security labels.

To further clear the referee’s concern, we have calculated the instrument costs related to surface washing, plasma cleaning and spin coating and the material cost for our security labels and added the results in Supplementary Note S1.

Instrument costs:

Plasma cleaning machine: US\$ 8000 (PDC-MG Mingheng company)

Spin coater: US\$ 500 (SC-1B Chuangshiweina company)

Ultrasonicator: US\$ 30 (J JP-010T Jiemeng cleaning equipment co. LTD)

These machines can last for ten years even longer. If we assume that only a 1,000,000 security labels are produced by these machines, the instrument cost for each security label

is less than US\$ 9×10^{-3} .

Material costs are shown in Supporting information Note S1

In terms of image characterization, it seems the referee misunderstand how the security labels are learnt by AI. We added the description in Methods as follows “For a typical process, one dot in lower-left representing a security label is captured as an image. Such a clear image was rotated by a step of 0.72° using an algorithm, obtaining a set including 500 training images.” This process generates 500 images and is done automatically by a computer. The rotated images are not stored after learning. A regular personal computer can do the job. Therefore, the cost for this can also be ignored.

To conclude, the overall cost of one unclonable inkjet-printed security label according to our strategy is about one cent (US\$ 11×10^{-3}).

REVIEWERS' COMMENTS:

Reviewer #1 (Remarks to the Author):

Dear Authors

Great work in revising the manuscript. In my opinion this is a valuable addition to the field that should be published in Nature Communications without further delay.

I cannot find any faults with the work at this stage. I would very much appreciate that the table listed in the response to referee #1, comment (15) is added to the supporting information. I also suggest including the newest PUF based system we recently published in ACS applied materials and interfaces.

I further urge the authors to stress that machine learning is a black box. Nobody knows how it works. This is an advantage as it is tamper proof, but it should be clearly stated that it works, but we do not know how it works.

Best wishes
Thomas Just Sørensen

Reviewer #2 (Remarks to the Author):

Newly submitted the manuscript is prepared in fully supported the responses of previous review comments.

Reviewer #3 (Remarks to the Author):

I feel that the authors have fully and satisfactorily addressed the concerns expressed by me and the other reviewers.

I appreciate the completeness of your responses.

REVIEWERS' COMMENTS:

Reviewer #1 (Remarks to the Author):

Dear Authors

Great work in revising the manuscript. In my opinion this is a valuable addition to the field that should be published in Nature Communications without further delay.

I cannot find any faults with the work at this stage. I would very much appreciate that the table listed in the response to referee #1, comment (15) is added to the supporting information. I also suggest including the newest PUF based system we recently published in ACS applied materials and interfaces.

I further urge the authors to stress that machine learning is a black box. Nobody knows how it works. This is an advantage as it is tamper proof, but it should be clearly stated that it works, but we do not know how it works.

Best wishes

Thomas Just Sørensen

Response: We thank the reviewer for the appreciation of our work. We have the table listed in the response to referee #1 in Supplementary Table 2 and cited the reference of ACS Appl. Mater. Interfaces 2019, 11, 6475–6482 accordingly.

We further added the “Deep learning, as a black box that nobody actually knows how it works in details up to now, is an advantage for unclonable anti-counterfeiting technique because it is tamper proof.”

Reviewer #2 (Remarks to the Author)

Newly submitted the manuscript is prepared in fully supported the responses of previous review comments.

Response: We thank the reviewer for acknowledging the completeness and significance of our work.

Reviewer #3 (Remarks to the Author):

I feel that the authors have fully and satisfactorily addressed the concerns expressed by me and the other reviewers.

I appreciate the completeness of your responses.

Response: We thank the reviewer for acknowledging the completeness and significance of our work.